# HyperPotter: Spell the Charm of High-Order Interactions in Audio Deepfake Detection

**Qing Wen** [1 2]   **Haohao Li** [1]   **Zhongjie Ba** [1 3]   **Peng Cheng** [1 3]   **Miao He** [1]   **Li Lu** [1 3]   **Kui Ren** [1 2]

## Abstract

Advances in AIGC technologies have enabled the synthesis of highly realistic audio deepfakes capable of deceiving human auditory perception. Although numerous audio deepfake detection (ADD) methods have been developed, most rely on local temporal/spectral features or pairwise relations, overlooking high-order interactions (HOIs). HOIs capture discriminative patterns that emerge from multiple feature components beyond their individual contributions. We propose HyperPotter, a hypergraph-based framework designed to capture high-order relations associated with synergistic patterns through clustering-based hyperedges with class-aware prototype initialization. Extensive experiments on 13 test sets show that HyperPotter improves over the baseline on 11 sets, yielding an average relative EER reduction of 12.68% across all test sets and 22.15% on the improved sets. These results demonstrate strong cross-scenario generalization, while also revealing robustness limits under severe codec or channel distortion.

## 1. Introduction

The emergence of AIGC enables highly realistic synthetic speech, supporting legitimate applications such as audiobooks while introducing serious security risks (Wang et al., 2025a). Recent reports indicate a surge in illicit activities using AI-generated voices (Lyu, 2025). In particular, voice forgery has been utilized to spread misinformation, commit identity fraud, and influence political campaigns (Brown,

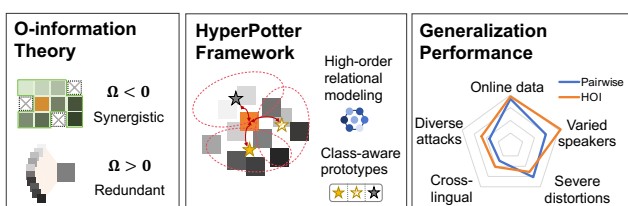

*Figure 1.* Motivated by O-information analysis, the HyperPotter framework enables high-order relation modeling and achieves competitive generalization results across most cross-scenario settings.

2025; NPR, 2025). As speech synthesis quality improves, artifacts distinguishing real and synthetic speech become increasingly subtle and distributed across multiple acoustic dimensions, posing challenges for effective detection.

To distinguish synthetic from authentic audio, extensive efforts have been devoted to audio deepfake detection (ADD). Existing ADD approaches rely on CNN-, graph-, or Transformer-based architectures to learn discriminative artifacts (Yang et al., 2025; Tak et al., 2021a; Jung et al., 2022; Dowerah et al., 2026). Despite their effectiveness, most approaches predominantly rely on **local or pairwise interactions** as basic relational units. For instance, convolutional operators focus on localized patterns, attention mechanisms compute pairwise token affinities, and graph-based models aggregate information along binary edges, limiting their ability to characterize joint behavior across multiple embeddings.

This motivates us to revisit audio deepfake detection from the perspective of high-order interactions. In complex systems, interactions involving more than two elements may exhibit behaviors that cannot always be reduced to pairwise dependencies, known as **high-order interactions (HOIs)** (Battiston et al., 2021). For ADD, this suggests a complementary view: some synthetic artifacts may not be reliably exposed by inspecting isolated features or pairwise similarities, but may become discriminative when multiple temporal-spectral cues are considered together.

To characterize such collective dependencies, we adopt a diagnostic tool — O-information (Rosas et al., 2019), a theoretical measure that distinguishes whether HOIs convey predominantly **synergistic** or **redundant** information. As

---

[1]The State Key Laboratory of Blockchain and Data Security, Zhejiang University, Hangzhou, China [2]Shanghai Institute for Advanced Study, Zhejiang University, Shanghai, China [3]Hangzhou High-Tech Zone (Binjiang) Institute of Blockchain and Data Security, Hangzhou, China. Correspondence to: Zhongjie Ba <zhongjieba@zju.edu.cn>, Peng Cheng <peng_cheng@zju.edu.cn>.

illustrated in Figure 1 (left), synergy captures information that is only available when multiple elements are considered jointly, whereas redundancy reflects information duplicated across them. Under this lens, we hypothesize that *generalizable spoofing artifacts contain high-order synergistic structures that should be explicitly modeled rather than implicitly approximated by pairwise mechanisms*.

To instantiate such high-order modeling, we propose a prototype-oriented hypergraph detection framework, namely **HyperPotter**. Unlike pairwise graphs that may redundantly encode similar evidence across edges, hypergraphs represent groups of temporal and spectral components as hyperedges, offering a natural structure for modeling synergistic HOIs. HyperPotter further incorporates two key components: 1) a relational artifact amplification module that emphasizes informative synergistic artifacts, and 2) a class-aware prototype-oriented hyperedge initialization mechanism for efficient relational construction. Extensive experiments on 13 evaluation sets show that HyperPotter improves over the Wav2Vec2-AASIST baseline on 11 sets and achieves competitive cross-scenario generalization (Figure 1 right[1]). Compared with current state-of-the-art methods, it further obtains an average relative EER reduction of 13.96% on 4 challenging datasets. These results highlight the value of HOI modeling for transferable spoofing artifacts, while the degradation under severe distortion suggests that such conditions may obscure fine-grained dependencies.

Our contributions are summarized as follows:

1. We provide the first investigation of high-order effects in audio deepfake detection from an information-theoretic perspective, using O-information as a diagnostic tool to probe redundancy- and synergy-dominated relational patterns.

2. We propose HyperPotter, a novel hypergraph-based detection framework that explicitly models multi-node relations via prototype-guided hyperedge construction and relational artifact enhancement.

3. We conduct extensive experiments on 13 diverse test sets, demonstrating improved cross-scenario generalization over the baseline while identifying limitations under severe codec and channel distortion.

Overall, this work underscores the importance of high-order interactions in audio deepfake detection and demonstrates that explicitly modeling synergistic dependencies offers an effective path toward improved generalization.

**Conflict of Interest Disclosure.** The authors declare no financial conflicts of interest related to this work.

---

[1]The comparison is supported by the statistics in Table 8.

## 2. Related Works

### 2.1. Audio Deepfake Detection

Early works used GMM (Patel & Patil, 2015) classifiers to process handcrafted features, such as MFCC, LFCC (Davis & Mermelstein, 1980), and CQCC (Todisco et al., 2017). With the development of deep learning, end-to-end models have become prevalent. For instance, RawNet2 (Tak et al., 2021a) constructs residual CNNs to capture the subtle details. RawGAT-ST (Tak et al., 2021b) and AASIST (Jung et al., 2022) transform waveforms into temporal-spectral heterogeneous graphs. Self-supervised learning (SSL) frontends (Wang & Yamagishi, 2022), usually comprising CNNs and Transformers, have leveraged large-scale speech knowledge to enhance detection robustness (Dowerah et al., 2026; Tak et al., 2022b; Rosello et al., 2023). However, these works primarily focus on local or pairwise relationships, overlooking the complex HOIs in heterogeneous artifacts.

### 2.2. Hypergraph

Hypergraph neural networks (Yang & Xu, 2025) have been widely used to model high-order relations in recommendation (Zhao et al., 2024), time series forecasting (Chen et al., 2025), and LLM enhancement (Huang et al., 2025). They are also effective in capturing complex multimodal dependencies. In computer vision, ViHGNN (Han et al., 2023) models correlations among image patches, and the subsequent works further improve performance through multi-scale hyperedges (Li et al., 2025) or virtual vertices (Fixelle, 2025). In the audio domain, LHGNN (Singh et al., 2025) achieves good performance in audio classification by combining local and high-order relations. However, we have not seen the exploration of hypergraph for ADD despite the importance of relational representation in this domain.

### 2.3. Prototype Mechanism

Prototype learning has been extensively studied for unsupervised domain adaptation, where class prototypes provide stable class-aware representations that address sampling variability and class imbalance (Tanwisuth et al., 2021) and are used to assign pseudo-labels (Pan et al., 2019). Recent advances extend this mechanism to anomaly detection via intrinsic normal prototypes extracted from test images (Luo et al., 2025) and incremental deepfake detection using domain-invariant clues and prototype similarity for replay selection (Tian et al., 2024; Pan et al., 2023). We leverage class-aware prototypes to initialize hyperedges, constructing semantically meaningful groupings that capture high-order interactions in audio features.

# 3. Analysis for HOIs in ADD

Existing ADD studies rarely analyze synthetic artifacts from the perspective of high-order interactions. In this section, we employ the O-information theory proposed by Rosas *et al.* (Rosas et al., 2019) as an interpretive tool to characterize HOI effects in the context of the ADD task. We first introduce key concepts to understand how HOIs behave differently in redundancy- versus synergy-dominated dependencies. This fundamental gap motivates us to explore high-order structures beyond pairwise networks and compare two different modeling strategies for ADD.

## 3.1. Information-theoretic Descriptions of HOIs

Let $\mathbf{X}^n = (X_1, \ldots, X_n)$ denote the set of $n$ random variables representing the feature space in ADD (e.g., multi-view representations). The uncertainty of the system is measured by the Shannon entropy $H(\mathbf{X}^n)$. The sum of the entropies for each individual variable is $\sum_{i=1}^n H(X_i)$, which is not less than $H(\mathbf{X}^n)$.

Following the description by Rosas *et al.*, HOIs are modeled as the multivariate interdependencies in the system. To quantify the interdependencies, we utilize two complementary metrics. First, *total correlation* or *multi-information* measures the degree of statistical dependence among multiple variables, defined as the sum of individual entropies minus the joint entropy:

$$C(\mathbf{X}^n) := \sum_{i=1}^n H(X_i) - H(\mathbf{X}^n),$$

A larger $C(\mathbf{X}^n)$ indicates stronger collective constraints that restrict group of variables. On the other hand, *dual total correlation* or *binding information* captures the extent to which each variable can be constrained or informed by the remaining variables:

$$B(\mathbf{X}^n) := H(\mathbf{X}^n) - \sum_{i=1}^n H(X_i|\mathbf{X}_{-i}^n),$$

where $\mathbf{X}_{-i}^n = (X_1, \ldots, X_{i-1}, X_{i+1}, \ldots, X_n)$. And the sum of $H(X_i|\mathbf{X}_{-i}^n)$ quantifies individual private randomness that cannot be inferred from observing other variables. Thus, $B(\mathbf{X}^n)$ quantifies information accessible only through the concurrent observation of multiple variables.

The *O-information* unifies these two metrics to determine the dominant nature of the multivariate interdependency:

$$\Omega(\mathbf{X}^n) := C(\mathbf{X}^n) - B(\mathbf{X}^n)$$
$$= (n-2)H(\mathbf{X}^n) + \sum_{i=1}^n [H(X_i) - H(\mathbf{X}_{-i}^n)].$$

The sign of $\Omega(\mathbf{X}^n)$ provides a useful diagnostic: $\Omega(\mathbf{X}^n) > 0$ suggests **redundancy-dominated** dependence (C > B), where variables tend to share overlapping information under a large set of constraints, providing robustness but low uniqueness; $\Omega(\mathbf{X}^n) < 0$ suggests **synergy-dominated** dependence (B > C), where information is more strongly expressed through joint observation of multiple variables. In this work, we use this interpretation to analyze learned representations rather than to claim a formal decomposition into pure redundancy or pure synergy.

## 3.2. A New Perspective Beyond Pairwise Interactions in ADD

In the context of an ADD system, the observable variables are audio embeddings extracted from a waveform sample. When most embeddings encode similar artifacts ($\Omega(\mathbf{X}^n) > 0$), identification results remain largely unchanged whether one considers individual embeddings or the entire embedding set. In contrast, when embeddings exhibit high randomness and diversity ($\Omega(\mathbf{X}^n) < 0$), reliable identification can only be achieved by modeling the synergistic structure of their interdependencies, instead of focusing on isolated parts.

Armed with the *O-information* framework, we analyze the underpinnings of current ADD approaches. Pairwise mechanisms (i.e., edges in standard graphs) are primarily suited to modeling redundancy-dominated interactions ($\Omega(\mathbf{X}^n) > 0$), where information is relatively repetitive across node pairs. Consequently, mainstream ADD methods, which mainly rely on pairwise mechanisms (e.g., Transformer or AASIST), tend to implicitly assume that deepfake artifacts can be sufficiently characterized through redundant feature correlations.

However, this assumption is challenged by empirical observations in recent literature. First, information bottleneck analyses suggest that removing task-irrelevant redundancy (corresponding to an increase in $B(\mathbf{X}^n)$) facilitates the extraction of fundamental acoustic features to improve detection performance (Eom et al., 2022; Ba et al., 2024). Second, it has been demonstrated that modeling interactions across disjoint frequency bands and temporal scales in heterogeneous networks leads to strong empirical performance (Jung et al., 2022). These observations can be interpreted as capturing the weak (i.e., the low correlations between features) but key constraints related to $C(\mathbf{X}^n)$.

Although a rigorous quantification of the synergy in ADD remains an open problem, these observations motivate a paradigm shift. We hypothesize that *advanced deepfake artifacts contain a fair amount of synergy information*. Therefore, we propose to examine whether ADD can benefit from modeling relations beyond pairs of embeddings. To achieve this, we introduce hypergraphs, which naturally capture high-order relationships in systems characterized by highly nonlinear interactions (Battiston et al., 2020), as the core

architecture for our method. This paradigm shift could open a new route for understanding deepfake artifacts, thereby improving detection generalization.

# 4. Hypergraph-based Detection Framework

## 4.1. Overview

HyperPotter formulates the ADD task as a graph-level classification problem and is built around a memory-enhanced hypergraph attention layer (HAGNN), as illustrated in Figure 2. The input waveforms are first processed by an encoding network to derive node features for hypergraph construction. For each graph, HAGNN constructs FCM-based hyperedges capturing HOI patterns, which are then strengthened by an attention-driven relational artifact amplification module. These hyperedges are initialized under the guidance of prototypes introduced in the next section.

## 4.2. Node Feature Encoding

To transform raw waveforms into node-level representations, we adopt the graph construction module from Wav2Vec2-AASIST, further described in Appendix A.2. Specifically, the input waveforms are encoded by the pretrained XLS-R model together with a RawNet2 encoder. The encoded embeddings are aggregated along spectral and temporal dimensions to form two sets of nodes. These node representations serve as inputs to subsequent HAGNN layers.

**Notation.** We now introduce the notation for node representations from the encoding process. Let $\mathcal{X}$ denote a batch of $B$ audio samples. For an audio sample $b \in \{1, \ldots, B\}$, we view its audio embeddings as a set of nodes and represent the sample as a hypergraph. Specifically, each sample is characterized by a node feature matrix $\mathbf{X}_b = [\mathbf{x}_{b,1}, \ldots, \mathbf{x}_{b,N}]^\top \in \mathbb{R}^{N \times D}$, where $N$ and $D$ represent the number of nodes and the feature dimension. The hypergraph contains $K$ soft hyperedges, and each node has a membership weight to each hyperedge. The number of "effective participation" nodes connected by a hyperedge is defined as cardinality $R$, which can be controlled by thresholding memberships. Each hypergraph is associated with a graph-level label $y_b \in \{0, 1\}$, consistent with the ground-truth label of the corresponding audio sample.

## 4.3. Hypergraph Attention Layer

The HAGNN models latent relational patterns by leveraging hypergraph-based message passing (Arya et al., 2020). It first aggregates node features into hyperedge representations via a FCM-based node-to-hyperedge mapping, and then propagates the resulting high-order relational information back to nodes through hyperedge-to-node message passing, thereby refining node representations. The overall process is

summarized as pseudocode in Algorithm 1 in the Appendix.

**Hyperedge Construction.** A hyperedge is defined as a non-empty cluster of nodes that jointly characterize high-order relations (Antelmi et al., 2023). Here, hyperedges are constructed through the Fuzzy C-Means (FCM) clustering method, which enables flexible modeling of node-hyperedge associations. Unlike hard clustering algorithms such as K-Means, FCM adopts a soft clustering strategy that allows each node to participate in multiple clusters to varying degrees. The soft clustering property makes FCM appropriate for modeling continuous associations in multimodal data (Li et al., 2025; Singh et al., 2025).

The FCM algorithm computes node feature distances to construct $K$ overlapping clusters, namely hyperedges. FCM optimizes the node-cluster assignments through an iterative process that alternately updates centroids and membership values. Each cluster is centered around a single centroid. Further, the membership matrix $\mathbf{U} \in \mathbb{R}^{N \times K}$ is defined as a soft incidence matrix, where each node can participate in multiple hyperedges with different strengths.

For a given graph sample, the node features are denoted by $\mathbf{X} = [\mathbf{x}_1, \cdots, \mathbf{x}_N]^\top$, where $\mathbf{x}_i \in \mathbb{R}^D$ represents the $D$-dimensional feature vector of the $i$-th node. For $K$ hyperedges, their cluster centroids are $\mathbf{C} \in \mathbb{R}^{K \times D}$. The membership degree of node $\mathbf{x}_i$ belonging to $k$-th hyperedge is represented by $u_{ik}$, where $0 \le u_{ik} \le 1$ and $\sum_{k=1}^{K} u_{ik} = 1$.

First, the membership matrix $\mathbf{U} \in \mathbb{R}^{N \times K}$ is initialized randomly. Then, the $k$-th centroid is computed as the membership-weighted average of node feature vectors:

$$\mathbf{c}_k = \frac{\sum_{i=1}^{N} (u_{ik})^m \mathbf{x}_i}{\sum_{i=1}^{N} (u_{ik})^m}, \tag{1}$$

where $m > 1$ is the fuzzifier parameter that controls the smoothness of $U$. When $m = 1$, FCM degenerates to the K-Means algorithm. Given updated centroids, the membership values are recomputed by:

$$u_{ij} = \frac{1}{\sum_{k=1}^{K} \left(\frac{d_{ij}}{d_{ik}}\right)^{\frac{2}{m-1}}}, \quad d_{ik} = \|\mathbf{x}_i - \mathbf{c}_k\| + \varepsilon, \tag{2}$$

where $d_{ik}$ denotes the distance between node feature vector $\boldsymbol{x}_i$ and centroid $\mathbf{c}_k$ and $\varepsilon$ is a small constant. A smaller distance $d_{ij}$ results in a larger membership value $u_{ij}$.

The iterative update process terminates, indicating that hyperedge representations have stabilized, when either the maximum number of iterations is reached or the loss function converges. The standard FCM objective function is: $J(\mathbf{U}, \mathbf{C}) = \sum_{i=1}^{N} \sum_{k=1}^{K} u_{ik}^m \|\boldsymbol{x}_i - \boldsymbol{c}_k\|^2$.

**Clustering Information Aggregation.** Following the node-to-hyperedge clustering, the hyperedge-level representations

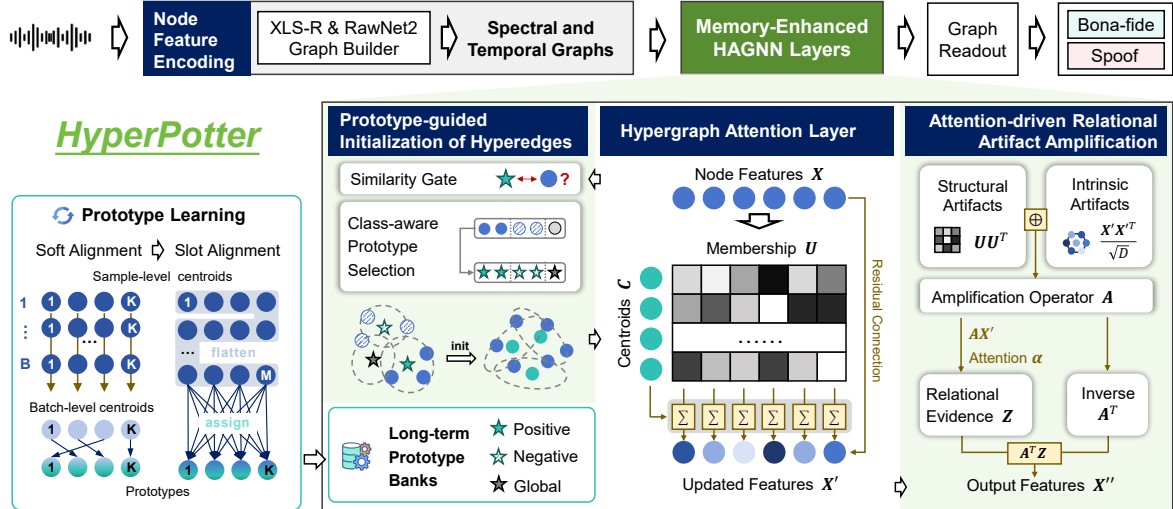

*Figure 2.* Overview of the HyperPotter framework. HyperPotter integrates hypergraph attention layers with prototype-guided hyperedge initialization to capture and amplify high-order relational artifacts, enabling effective aggregation of discriminative cues for generalizable audio spoofing detection.

are propagated back to nodes. Specifically, the hyperedge-to-node aggregation process updates node features via a residual fusion mechanism: $\mathbf{x}'_i = \beta_1 \mathbf{x}_i + (1 - \beta_1) \sum_{k=1}^{K} u_{ik} \mathbf{c}_k$, where $\beta_1$ balances the contribution of original features and the aggregated information from hyperedges.

### 4.4. Relational Artifact Amplification

**Artifact Amplification Operator.** To adapt the model for deepfake detection, we propose an operator designed to amplify the relational artifacts within the hypergraph and node embeddings. The operator fuses a structural term and a feature term: $\mathbf{A} = \text{softmax}(\beta_2 \mathbf{A}^{(c)} + (1 - \beta_2) \mathbf{A}^{(f)})$, where $\beta_2$ balances the two components and the softmax is applied row-wise. Specifically, the structural term $\mathbf{A}^{(c)} = \mathbf{U}\mathbf{U}^{\mathsf{T}}$ captures shared hyperedge participation (whether nodes belong to the same high-order groups), while the feature term $\mathbf{A}^{(f)} = \frac{\mathbf{X}'\mathbf{X}'^{\mathsf{T}}}{\sqrt{D}}$ represents pairwise feature similarity. Under this formulation, the connectivity between two nodes is amplified if they exhibit both similar structural consistency and feature-level agreement. Notably, although $\mathbf{A}$ has an $N \times N$ pairwise form for efficient propagation, it is computed from hypergraph-derived quantities ($\mathbf{U}$ and $\mathbf{X}'$), and therefore summarizes shared participation in higher-order groups rather than raw pairwise similarity alone.

**Attention-driven Amplification.** Based on the operator $A$, we further introduce an attention mechanism to model manipulation traces via an aggregate-then-project-back update. First, relational evidence is aggregated as $\mathbf{Z} = \mathbf{A}\mathbf{X}'$. Next, it is reweighted by $\mathbf{Z}' = ((1 + \boldsymbol{\alpha}) \odot \mathbf{Z})$, where $\boldsymbol{\alpha} = \text{softmax}(\mathbf{Z}\mathbf{w}_{\alpha})$ is a node-wise attention factor and $\mathbf{w}_{\alpha}$ is learnable. Finally, the reweighted evidence is projected back into the node space: $\mathbf{X}'' = \mathbf{A}^{\mathsf{T}}\mathbf{Z}'$. (Since $\mathbf{A}$ is row-

normalized and thus directional, $\mathbf{A}^{\mathsf{T}}$ is used to distribute the reweighted evidence back along the reverse direction.) This mechanism encourages the model to focus on the most informative relational artifacts.

## 5. Prototype Bank Design

While the previous section details the hypergraph construction and the feature amplification, effective feature aggregation depends heavily on high-quality initialization. As illustrated in the left panel of Figure 2, we introduce the "Prototype-guided Initialization" for FCM clustering, equipping hypergraph layers with long-term structural memory. The prototype bank module stores multiclass centroids for the FCM initialization process to speed convergence and improve detection performance.

### 5.1. Motivation

**Problems of Randomly Initialized Hyperedges.** The default initialization method of the FCM-based hyperedge generates the membership matrix randomly. Specifically, each batch constructs HOIs from scratch, which strongly affects the efficiency and quality of relationship construction.

**Prototypes as Cross-batch Structural Priors.** To facilitate hyperedge initialization, we incorporate historical structural knowledge through a long-term storage mechanism. Inspired by the semantic graph search scheme (Fu et al., 2017), we compress information from hypergraph centroids into prototypes, which serve as semantic anchors. Compared to vector databases (Kang et al., 2024) or anchor graphs (Ju et al., 2024), prototypes offer advantages in capturing global high-order concepts and enhancing compu-

tational efficiency. Furthermore, these prototypes are updated via a non-differentiable Exponential Moving Average (EMA), which significantly improves optimization stability.

## 5.2. Prototype-Guided Hyperedge Initialization

**Components of a Prototype Bank.** To better capture artifact characteristics, we maintain a prototype bank that captures both class-aware discriminative features and global structural information. Formally, the bank is defined as $\mathcal{P} = [\mathbf{P}^{(+)}, \mathbf{P}^{(-)}, \mathbf{P}^{(g)}]$, where each set $\mathbf{P} = \{\mathbf{p}_k\}_{k=1}^{K}$ contains $K$ centroid-sized prototypes $\mathbf{p}_k \in \mathbb{R}^D$. Specifically, the class-conditional sets $\mathbf{P}^{(+)}$ and $\mathbf{P}^{(-)}$ leverage binary supervision to guide the clustering process, while the global set $\mathbf{P}^{(g)}$ serves as an alignment anchor to ensure overall feature consistency.

**Prototype-guided Hyperedge Initialization Process.** The process includes three sequential steps to refine the starting state of the FCM clustering: **1) Similarity gating.** To restrict domain matching to established prototypes, a similarity controller first evaluates each batch. The similarity score $s$ is defined as the cosine similarity between the batch-averaged features and the mean global prototypes: $s = \cos(\frac{1}{BN} \sum_{b=1}^{B} \sum_{n=1}^{N} \mathbf{x}_{b,n}, \frac{1}{K} \sum_{k=1}^{K} \mathbf{p}_k^{(g)})$. Batches with $s < \tau$, where $\tau$ is a predefined threshold, are filtered out to ensure feature alignment. **2) Prototype selection.** We then adaptively select prototypes based on label availability, which varies between stages. Evaluation: Initialization centroids are directly inherited from $\mathbf{P}^{(g)}$. Training: We employ a class-aware initialization strategy to construct $K$ centroids. The first 20% of the K slots are occupied by $\mathbf{P}^{(g)}$ to keep structural stability. The remaining slots are filled by randomly sampling $\mathbf{P}^{(+)}$ and $\mathbf{P}^{(-)}$, proportional to the ratio of bona-fide to spoof samples within the batch. **3) External centroid injection.** The externally constructed centroids, augmented with a minor perturbation, are then injected into the FCM algorithm. These centroids define the initial node membership, which triggers the subsequent FCM optimization loops.

## 5.3. Prototype Learning Pipeline

We introduce a dynamic update mechanism that integrates batch knowledge into the prototype bank. This process extracts class-aware and global centroids from the batch, aligns these local centroids with historical prototypes, and performs an EMA-based integration.

**Label-aware Centroids Estimation.** To integrate supervised information, we perform an additional FCM iteration that updates class-aware batch centroids with membership values weighted by node labels. For brevity, we describe the formulation using positive prototypes as the representative case. For $b$-th sample, the class-conditional membership

between $n$-th node and $k$-th cluster is $w_{b,nk}^{(+)} = (y_b \cdot u_{b,nk})^m$, where $m$ is the fuzziness. Following the equation 1, this $k$-th positive centroid for this sample is computed as:

$$\hat{\mathbf{c}}_{b,k}^{(+)} = \frac{\sum_{n=1}^{N} w_{b,nk}^{(+)} \mathbf{x}_{b,n}}{\sum_{n=1}^{N} w_{b,nk}^{(+)} + \varepsilon}. \tag{3}$$

**Semi-alignment Strategy.** Since local clusters from the current batch may be misaligned with the features of global prototype banks, we adopt a progressive two-stage alignment strategy: a soft-alignment phase for batch-level consistency in early epochs, and a slot-alignment phase for precise feature aggregation (improving the stability and specificity of prototype updates).

**Soft Alignment** facilitates the aggregation of irregular clusters in a batch. We compute batch-level centroids first, then update prototypes. Specifically, while the global batch centroid $\hat{\mathbf{c}}_k^{(g)}$ is the mean of all batch centroids, the class-aware batch centroid $\hat{\mathbf{c}}_k^{(+)}$ incorporates sample-level contributions. For class-aware batch centroids, we determine the contribution of each sample for cluster $k$ by defining a weight as $\alpha_{b,k}^{(+)} = \sum_{n=1}^{N} w_{b,nk}^{(+)}$. The $k$-th global and positive batch centroids are updated as follows:

$$\hat{\mathbf{c}}_k^{(g)} = \frac{1}{B} \sum_{b=1}^{B} \mathbf{c}_{b,k}, \quad \hat{\mathbf{c}}_k^{(+)} = \frac{\sum_{b=1}^{B} \alpha_{b,k}^{(+)} \hat{\mathbf{c}}_{b,k}^{(+)}}{\sum_{b=1}^{B} \alpha_{b,k}^{(+)} + \varepsilon}. \tag{4}$$

Then the soft alignment strategy aligns the batch-level centroids with long-term prototypes. We compute a cosine-similarity matrix $S_{ij} = \cos(\mathbf{p}_i^{(g)}, \hat{\mathbf{c}}_j^{(g)})$. Starting from $I_0 = J_0 = \varnothing$, at iteration $t$ we select $(i_t, j_t) = \arg\max_{i \notin I_{t-1}, j \notin J_{t-1}} S_{ij}$, update $I_t = I_{t-1} \cup i_t$ and $J_t = J_{t-1} \cup j_t$, and obtain a greedy assignment $\pi_{\text{greedy}}(j_t) = i_t$. We then reorder centroids as $\tilde{c}_j = c_{\pi_{\text{greedy}}(j)}$. The same permutation is used for both positive and negative centroids.

**Slot Alignment** enables every centroid to contribute to the corresponding prototype slots in a direct and precise manner. First, centroids from all batches are flattened into a sequence $\mathcal{C} = \{\mathbf{c}_m\}_{m=1}^{M}$, where $M = B \cdot K$. To align the flattened centroids with the prototypes, we compute assignment weights as $a_{mk} = \text{softmax}(\cos(\mathbf{c}_m^{(g)} \cdot \mathbf{p}_k^{(g)}))$. Based on these assignments, the $k$-th batch-level global centroid is constructed as:

$$\tilde{\mathbf{c}}_k^{(g)} = \frac{\sum_{m=1}^{M} a_{mk} \mathbf{c}_m}{\sum_{m=1}^{M} a_{mk}}$$

The resulting slotted centroids are then fused with the average centroids $\hat{\mathbf{c}}_k^{(g)}$ (as defined in equation 4): $\tilde{\mathbf{c}}_k^{(g)} = (\tilde{\mathbf{c}}_k^{(g)} + \hat{\mathbf{c}}_k^{(g)})/2$ to improve robustness.

**EMA-based Update Mechanism.** To maintain cross-batch stability, the class-conditional prototypes are updated via

*Table 1.* Comparison of EERs (%) for different models trained on ASVspoof2019 LA. Lower is better. **Bold**, underlined, and † indicate the best, second-best, and third-best results, respectively. Subscripts denote the performance gaps between HyperPotter and its baseline.

| Model | Params (M) | In-the-Wild | ASV20 19LA | ASV20 21LA | ASV20 21DF | ASV spoof5 | FoR | Codecfake | ADD22 Track1 | ADD22 Track3 | ADD23 R1 | ADD23 R2 | Libri Voc | SONAR |
|---|---|---|---|---|---|---|---|---|---|---|---|---|---|---|
| Wav2Vec2+AASIST (Tak et al., 2022b) | 317.84 | 11.19 | 0.221 | 0.82 | 6.63 | 16.24 | 7.46 | 43.36 | 31.04 | 16.52 | 27.74 | 21.92 | 11.13 | 46.12 |
| XLSR+Conformer (Rosello et al., 2023) | 302.40 | - | - | 0.97 | 2.58 | - | - | 36.01 | - | - | - | - | - | - |
| XLSR+Conformer+TCM (Truong et al., 2024) | 319 | 7.79 | $0.19^\dagger$ | 3.00 | 2.15 | 18.85 | 10.69 | 36.01 | 37.40 | 20.94 | 23.43 | 22.74 | 2.35 | $26.57^\dagger$ |
| XLSR+AASIST2 (Zhang et al., 2024b) | - | - | 0.15 | 1.61 | 2.77 | - | - | - | - | - | - | - | - | - |
| WavLM+MFA (Guo et al., 2024) | - | - | 0.42 | 5.08 | 2.56 | - | - | - | - | - | - | - | - | - |
| XLSR+SLS (Zhang et al., 2024a) | 340 | $7.45^\dagger$ | 0.231 | 2.86 | 1.91 | 18.76 | $5.07^\dagger$ | **33.43** | 33.95 | 15.74 | **19.37** | 21.09 | **1.72** | 24.72 |
| XLSR+MoE (Wang et al., 2025b) | - | - | 0.74 | 2.96 | 2.54 | - | - | - | - | - | - | - | - | - |
| XLS-R+STCA+LMDC (Hao et al., 2025) | - | - | **0.09** | **0.78** | 1.87 | - | - | - | - | - | - | - | - | - |
| XLSR+Mamba (Xiao & Das, 2025) | 319 | 6.70 | 0.421 | $0.93^\dagger$ | $1.88^\dagger$ | $14.40^\dagger$ | 6.71 | $35.26^\dagger$ | 34.22 | 19.36 | 21.84 | **20.15** | $2.15^\dagger$ | **24.26** |
| XLSR+BiCrossMamba (El Kheir et al., 2025) | 318.21 | 7.94 | 0.71 | 3.83 | 2.35 | 13.67 | 6.85 | 37.70 | **30.44** | 18.69 | 29.44 | 29.93 | 2.12 | 27.36 |
| Wav2Vec2+VIB(w/ HAR) (Doan et al., 2024) | - | 3.78 | 2.68 | - | 2.07 | - | 10.32 | - | - | - | - | - | - | - |
| Wav2Vec2+VIB(w/o HAR) (Doan et al., 2024) | - | 9.78 | 0.57 | - | 3.38 | - | - | - | - | - | - | - | - | - |
| **Wav2Vec2+AASIST(Baseline)** | 317.84 | 7.58 | 0.26 | 2.48 | 4.08 | **13.38** | 4.24 | 40.22 | 33.79 | 16.14 | 25.55 | 22.21 | 6.96 | 32.02 |
| **HyperPotter** | 317.87 | $\mathbf{5.72}_{\downarrow1.86}$ | $0.23_{\downarrow0.03}$ | $3.94_{\uparrow1.46}$ | $\mathbf{1.78}_{\downarrow2.30}$ | $16.04_{\uparrow2.66}$ | $\mathbf{3.89}_{\downarrow0.35}$ | $34.47_{\downarrow5.75}$ | $32.34^\dagger_{\downarrow1.45}$ | $\mathbf{11.31}_{\downarrow4.83}$ | $21.49_{\downarrow4.06}$ | $21.84^\dagger_{\downarrow0.37}$ | $2.55_{\downarrow4.41}$ | $27.71_{\downarrow4.31}$ |

*Note:* HAR module of Wav2Vec2+VIB relies on re-synthesized augmentation (i.e., extra generated training data); therefore, we treat the w/o HAR variant as the primary fair comparison.

an EMA: $\hat{\mathbf{P}}^{(+)} = \mu\tilde{\mathbf{C}}^{(+)} + (1-\mu)\mathbf{P}^{(+)}$, where $\mu$ denotes the momentum coefficient. The final global prototypes are computed as: $\hat{\mathbf{P}}^{(g)} = \mu(\gamma\tilde{\mathbf{C}}^{(g)} + (1-\gamma)\mathbf{N}) + (1-\mu)\mathbf{P}^{(g)}$. Here, $\mathbf{N} = 0.5(\hat{\mathbf{P}}^{(+)} + \hat{\mathbf{P}}^{(-)})$ represents a class-unbiased neural prototype. This formulation effectively fuses local structural cues with unbiased long-term anchors.

# 6. Experiments

In this section, we evaluate the generalization performance of the HyperPotter framework across multiple datasets and demonstrate that HOIs are effectively captured and leveraged to describe synthetic artifacts across diverse scenarios.

## 6.1. Experimental Setup

**Datasets and Evaluation Metrics.** All models are trained only on the ASVspoof2019 LA training set. Following the evaluation protocol of the Speech DF Arena (Dowerah et al., 2026), we evaluate model performance on 13 test sets. These datasets are categorized by language, source dataset, codec, and spoofing method, with detailed descriptions provided in Appendix C. We report Equal Error Rate (EER) and F1 score as evaluation metrics.

**Implementation Details.** Wav2Vec2-AASIST is used as the backbone, with its graph builder and readout mechanism preserved. Guided by prototype banks, two HAGNN layers are applied to spectral and temporal graphs, respectively. Similar to the HS-GALs in AASIST, we implement four heterogeneous HAGNN layers to model spectral-temporal heterogeneous graphs.

A warm-start mechanism is employed during training to control prototype learning and usage. K-Means is used to initialize FCM and is replaced with prototypes at the 5*th* epoch. At the first training batch, all prototypes are initialized with the mean of centroids ($\hat{\mathbf{c}}_k^{(g)}$ in Eq. 4). Prototypes are updated using the soft alignment before the 20*th* epoch, and followed by the slot alignment algorithm in later stages. Starting at epoch 5, the similarity gate $\tau$

decays linearly from 0.1 to 0. During evaluation, prototypes remain fixed to avoid information leakage. Other implementation settings are provided in Appendix B.1. The code and pre-trained models are available at https://github.com/HyperPotter/HyperPotter.

## 6.2. Generalization Performance

**Comparison with SOTAs.** To ensure a fair comparison, we evaluate HyperPotter against current SOTA models with comparable model sizes (approximately 300M-400M parameters). We also retrain the baseline model, Wav2Vec2-AASIST, under the same training settings as HyperPotter to assess the performance gains accurately. The EER and F1-score results are shown in Table 1 and Table 6. HyperPotter achieves the best EER on several challenging datasets, including In-the-Wild (5.72%), ASVspoof2021 DF (1.78%), FoR (3.89%), and ADD2022 Track3 (11.31%). These datasets represent diverse speakers, varied spoofing attacks, and challenging real-world, cross-lingual conditions. On the remaining datasets, HyperPotter also demonstrates competitive performance. With only 0.03M additional parameters, HyperPotter outperforms the baseline on most datasets. Moderate performance degradation is observed on the 2021LA and ASVspoof5 datasets, where speech samples are heavily affected by diverse codec-based compression and transmission processes. We hypothesize that such distortions act as "masks" that obscure high-order interdependences across the temporal and spectral domains. To enhance HyperPotter under severe channel interference, we explore different codec augmentation strategies, with results shown in Table 11. Overall, these results suggest that while severe channel interference can mask high-order relational structures, HyperPotter excels in scenarios where speech signals maintain structural integrity. This distinctive specialization positions HyperPotter as a critical "specialist" in modern anti-spoofing ecosystems. Such a role aligns well with the ongoing shift in practical ADD systems toward ensemble or Multi-Expert (MoE) frameworks for diverse deployment conditions.

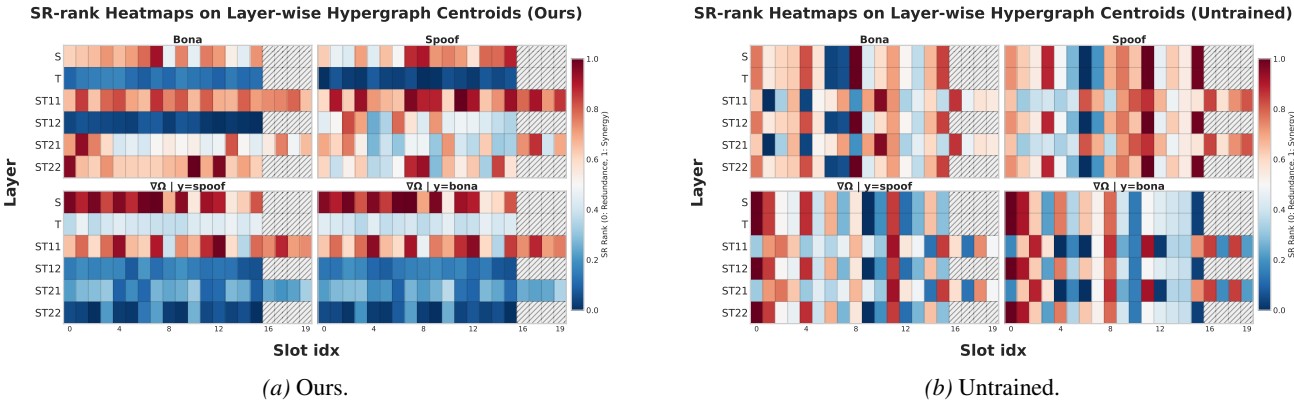

*(a)* Ours.         *(b)* Untrained.

*Figure 3.* Synergy-Redundancy rank heatmaps for global-prototype-aligned centroid slots across hypergraph layers: O-information ($\Omega$) for spoof/bona-fide data and Gradient O-information conditioned on label $y$ ($\nabla\Omega \mid y$), with SR min–max normalized to $[0, 1]$. High (red) values indicate that the centroid's interactions are predominantly synergistic, while low (blue) values indicate they are primarily redundant.

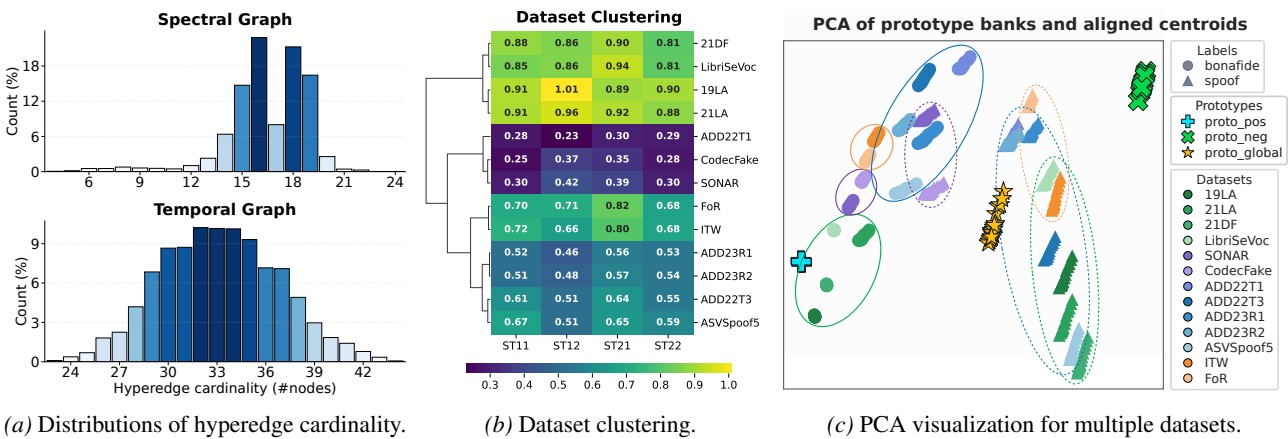

*(a)* Distributions of hyperedge cardinality.     *(b)* Dataset clustering.     *(c)* PCA visualization for multiple datasets.

*Figure 4.* Visualizations demonstrating the effectiveness of HOI modeling and its capability to describe synthetic artifacts.

### 6.3. Representation Capability for High-order Relations

**HOI Distribution Visualization.** We analyze the redundancy-synergy patterns of learned hyperedge representations by computing O-information metrics on layer-wise hyperedge centroids. Figure 3 visualizes the resulting synergy-redundancy ranks and compares HyperPotter with an architecture-matched untrained model. The trained model exhibits more structured centroid-level patterns across layers and classes, indicating that the hypergraph module learns nontrivial high-order relational structure. These results complement the generalization experiments and support the usefulness of high-order relational modeling in ADD. More details are provided in Appendix B.3.

**Distribution of Hyperedge Cardinality.** To provide evidence for the presence of HOIs, we analyze the distribution of hyperedge cardinalities, the number of nodes connected by a hyperedge, followed by a hyperedge pruning with effective membership values larger than $1/\mathcal{R}$ to ensure a clear

and meaningful observation. Figure 4a illustrates the results for the spectral and temporal HAGNN blocks, which consist of 42 and 66 total nodes, respectively. We observe that most cardinalities are concentrated around the median node count, indicating HyperPotter tends to capture high-order connections involving multiple nodes selectively rather than relying solely on pairwise interactions (cardinality of 2).

**Data Clustering based on Relational Connections.** To evaluate the synthetic discriminability of HOIs, we investigate relational similarity patterns based on hyperedges at both dataset and sample levels. We extract HAGNN centroids from test samples and compute cosine similarities to positive and negative prototypes of bona-fide and spoof classes, from which sample–prototype distribution gaps are derived for each evaluation dataset. Inter-dataset relational similarities are visualized using a clustermap (Figure 4b), where hierarchical clustering across four HAGNN layers groups datasets with similar patterns closer together. The resulting clustering structure is consistent with our analysis of dataset generation mechanisms in Table 5. At the embed-

ding level, we visualize the prototype banks and slot-aligned centroids using PCA (Figure 4c), where datasets are categorized by data source: traditional deepfake data (green), neural codec data (purple), complex and unspecified data (blue), and online data (orange). Solid and dashed lines denote bona-fide and spoof data, respectively. We observe that bona-fide centroids from similar sources exhibit close distribution, while spoofing centroids from different data types are generally clustered in a compact region (with the exception of codec data[2]). This suggests that high-order relational modeling effectively captures shared spoofing characteristics across diverse datasets. More details about our metric and sample-level analyses are provided in the Appendix F.5.

## 6.4. Ablation Studies

**Influence of HyperPotter Sub-modules.** We conduct an ablation study on the sub-modules of HyperPotter and alignment algorithms, as shown in Table 2 and Table 9. Performance degradation is observed when key components are removed, especially under real-world conditions (ITW and FoR) and diverse spoofing attacks (21DF), demonstrating the importance of these modules in effective HOI modeling. However, the benefits are not uniform under severe codec/channel distortion, where removing the amplification module or prototype banks can improve 21LA or ASVspoof5 performance. This suggests that codec-heavy conditions may favor more redundancy-oriented cues. The convergence behavior of different models are shown in Appendix D.4. Although the inclusion of HAGNN blocks slows down convergence, introducing prototype banks alleviates this issue and improves training efficiency.

*Table 2.* Ablation study on HyperPotter sub-modules and alignment algorithms, reported as EERs (%).

| | ITW | 21LA | 21DF | ASV5 | FoR |
|---|---|---|---|---|---|
| **HyperPotter** | **5.72** | 3.94 | **1.78** | 16.04 | **3.89** |
| w/o Amplification Module | 6.9 ↑ | 3.50 ↓ | 1.88 ↑ | **14.46** ↓ | 6.45 ↑ |
| w/o Prototype Banks | 6.49 ↑ | **2.80** ↓ | 1.88 ↓ | 15.42 ↓ | 4.59 ↑ |
| w/o Both Alignment Algs | 5.95 ↑ | 3.83 ↓ | 2.25 ↑ | 19.07 ↑ | 4.46 ↑ |
| w/o Soft Alignment Alg | 7.45 ↑ | 3.25 ↓ | 2.05 ↑ | 19.42 ↑ | 4.02 ↑ |
| w/o Slot Alignment Alg | 6.18 ↑ | 3.24 ↓ | 2.07 ↑ | 19.90 ↑ | 4.02 ↑ |

**Influence of Hyperedge Counts and Cardinality.** The number and cardinality of hyperedges, corresponding to the FCM cluster count and membership degree, directly determine the hypergraph structure. The results are reported in Table 3, indicating that using hyperedge counts of $25\%N$–$50\%N$ without explicit cardinality thresholding provides a reasonable trade-off. Smaller $\mathcal{R}$ imposes overly strict constraints on HOIs, delivering superior results un-

---

[2]The role of neural codec methods is not clearly defined, as it may be viewed either as augmentation methods in ASVspoof5 or as spoofing mechanisms in CodecFake.

der advantageous conditions, while $\mathcal{R} = 2$ reduces the hypergraph to a pairwise graph and leads to performance degradation. These results underscore the necessity of HOIs, especially in enhancing ADD performance under complex scenarios characterized by diverse spoofing algorithms and multiple speakers.

*Table 3.* Comparison of different hyperedge counts ($K$) and cardinality ($\mathcal{R}$), evaluated using EER (%). $N$ denotes the number of nodes in the input graph.

| | ITW | 21LA | 21DF | ASVspoof5 | FoR |
|---|---|---|---|---|---|
| **HyperPotter** | 5.72 | 3.94 | 1.78 | **16.04** | 3.89 |
| $K = 0.25N$ | 5.88↑ | 3.63↓ | 1.92↑ | 18.23↑ | 4.06↑ |
| $K = 0.50N$ | 6.39↑ | 2.87↓ | 1.76↓ | 19.00↑ | 4.77↑ |
| $K = 0.75N$ | 6.09↑ | 3.37↓ | 1.76↓ | 19.22↑ | 4.86↑ |
| $\mathcal{R} = 0.25N$ | **4.97**↓ | 3.44↓ | **1.60**↓ | 18.89↑ | 5.61↑ |
| $\mathcal{R} = 0.50N$ | 5.97↑ | 2.71↓ | 2.04↑ | 17.74↑ | 3.93↑ |
| $\mathcal{R} = 0.75N$ | 6.02↑ | **2.44**↓ | 1.96↑ | 16.91↑ | **3.75**↓ |
| $\mathcal{R} = 2$ | 6.95↑ | 3.79↓ | 2.20↑ | 20.04↑ | 5.21↑ |

## 7. Discussion

**Non-uniform gains under codec/distortion.** HyperPotter improves generalization on most evaluation sets, but its gains are not uniform under severe codec or channel distortion. A possible reason is that heavy compression and transmission artifacts can mask fine-grained temporal-spectral relations, making redundancy-oriented cues more stable than synergy-emphasizing cues. Future work may explore adaptive mechanisms that balance redundancy- and synergy-oriented evidence according to channel conditions.

**Practical graph construction.** As shown in Table 13, HyperPotter introduces moderate computational overhead. While the current RawNet2-based graph builder provides a reliable foundation, lightweight graph construction modules (Fixelle, 2025) and graph augmentation strategies (Dong et al., 2025) may further improve deployment efficiency and robustness under channel variation and distortion.

## 8. Conclusion

This paper investigates audio deepfake detection from a high-order interaction perspective. Using O-information as a diagnostic tool, we provide suggestive evidence that learned hyperedge representations contain structured redundancy-synergy patterns. We propose HyperPotter, a prototype-oriented hypergraph framework for modeling multi-node relations and enhancing relational artifacts. Experiments on 13 datasets show improvements over the baseline on 11 datasets, with an average relative EER reduction of 12.68% across all datasets and 22.15% on the improved subsets, while also revealing limitations under severe codec/channel distortion.

## Acknowledgements

This work was supported by the Shanghai Municipal Special Program for Basic Research on General AI Foundation Models (Grant No. 2025SHZDZX025G17), in collaboration with Shanghai Artificial Intelligence Laboratory. It was also supported by the Zhejiang Provincial Natural Science Foundation of China (No. LD24F020010 and No. LY24F020007), the National Natural Science Foundation of China (No. 62472372 and No. 62572424) and the Key Research and Development Program of Hangzhou City (2024SZD1A27). We thank the authors of Wav2Vec2-AASIST for releasing their implementation, which served as the backbone for our experiments.

## Impact Statement

This paper presents research on deepfake audio detection, with the goal of improving the robustness and security of speech-based systems. By enhancing the ability to distinguish bona-fide from spoofed audio, this work may contribute to mitigating risks associated with malicious voice synthesis, such as fraud and impersonation. While audio analysis technologies can raise ethical considerations depending on their deployment, these concerns are not specific to this work and largely depend on downstream applications.

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

# A. Supplementary Explanation for HyperPotter Methodology

## A.1. Pseudocode in Sec. 4.3

Algorithm 1 summarizes the complete computation of the HAGNN layer. The procedure consists of three stages: prototype- or k-means++-based centroid initialization, FCM-based soft hyperedge construction, and relational artifact amplification.

---

**Algorithm 1** Hypergraph Attention Layer with FCM-based Hyperedge Construction

---

**Require:** Node features $\mathbf{X} \in \mathbb{R}^{B \times N \times d_{\text{in}}}$, number of clusters $K$, fuzziness $m$, maximum FCM iterations $T$, temperature $\tau$, dropout rate $p$

**Ensure:** Updated node features $\mathbf{Y} \in \mathbb{R}^{B \times N \times d_{\text{out}}}$

1: $K_{\text{eff}} \leftarrow \min(K, N)$
2: $\boldsymbol{\mu}_X \leftarrow \text{Mean}(\mathbf{X}, \dim = 1)$, $\boldsymbol{\sigma}_X \leftarrow \text{Std}(\mathbf{X}, \dim = 1) + \epsilon$
3: $\widehat{\mathbf{X}} \leftarrow (\mathbf{X} - \boldsymbol{\mu}_X)/\boldsymbol{\sigma}_X$
4: **if** external prototype centroids are available **then**
5:     Initialize $\widehat{\mathbf{C}} \in \mathbb{R}^{B \times K_{\text{eff}} \times d_{\text{in}}}$ from external prototypes
6:     Normalize $\widehat{\mathbf{C}}$ using $\boldsymbol{\mu}_X$ and $\boldsymbol{\sigma}_X$
7: **else**
8:     Initialize $\widehat{\mathbf{C}}$ from $\widehat{\mathbf{X}}$ using k-means++
9:     Use random initialization as fallback
10: **end if**
11: **for** $t = 1, \ldots, T$ **do**
12:     Compute pairwise distances: $\mathbf{D}_{bik} \leftarrow \|\widehat{\mathbf{X}}_{bi} - \widehat{\mathbf{C}}_{bk}\|_2 + \epsilon$
13:     Compute fuzzy memberships: $\mathbf{U}_{bik} \leftarrow \dfrac{(\mathbf{D}_{bik})^{-2/(m-1)}}{\sum_{j=1}^{K_{\text{eff}}} (\mathbf{D}_{bij})^{-2/(m-1)} + \epsilon}$
14:     Update centroids: $\widehat{\mathbf{C}}_{bk} \leftarrow \dfrac{\sum_{i=1}^{N} \mathbf{U}_{bik}^m \widehat{\mathbf{X}}_{bi}}{\sum_{i=1}^{N} \mathbf{U}_{bik}^m + \epsilon}$
15:     **if** the average centroid shift is smaller than $\epsilon$ **then**
16:         **break**
17:     **end if**
18: **end for**
19: $\mathbf{C} \leftarrow \widehat{\mathbf{C}} \odot \boldsymbol{\sigma}_X + \boldsymbol{\mu}_X$
20: $\mathbf{Z} \leftarrow \mathbf{U}\mathbf{C}$
21: $\widetilde{\mathbf{X}} \leftarrow 0.9\mathbf{X} + 0.1\mathbf{Z}$
22: $\mathbf{H} \leftarrow W_n\widetilde{\mathbf{X}}$, $\mathbf{H}_c \leftarrow W_c\widetilde{\mathbf{X}}$
23: $\overline{\mathbf{U}} \leftarrow \ell_2\text{-Normalize}(\mathbf{U}, \dim = -1)$
24: $\mathbf{S}_c \leftarrow \overline{\mathbf{U}}\,\overline{\mathbf{U}}^\top$
25: $\mathbf{S}_f \leftarrow \mathbf{H}\mathbf{H}^\top/\sqrt{d_{\text{out}}}$
26: $\mathbf{A} \leftarrow \text{Softmax}((0.6\mathbf{S}_c + 0.4\mathbf{S}_f)/\tau, \dim = -1)$
27: $\mathbf{R} \leftarrow \mathbf{A}\mathbf{H}$
28: **if** lightweight attention is enabled **then**
29:     $\boldsymbol{\alpha} \leftarrow \text{Softmax}\left(\text{Linear}_a(\mathbf{R})\mathbf{w}_a/\tau, \dim = -1\right)$
30:     $\mathbf{R} \leftarrow \mathbf{R} + \boldsymbol{\alpha} \odot \mathbf{R}$
31: **end if**
32: $\mathbf{O} \leftarrow \text{Dropout}(\mathbf{A}^\top\mathbf{R}, p)$
33: $\mathbf{G} \leftarrow \sigma(W_g[\mathbf{O}; \mathbf{H}_c])$
34: $\mathbf{F} \leftarrow \mathbf{G} \odot \mathbf{O} + (1 - \mathbf{G}) \odot \mathbf{H}_c$
35: $\mathbf{Y} \leftarrow \mathbf{F} + W_n\mathbf{X}$
36: $\mathbf{Y} \leftarrow \text{SELU}(\text{LayerNorm}(\mathbf{Y}))$
37: **return** $\mathbf{Y}$

---

## A.2. Baselines: AASIST-related Networks

AASIST (Jung et al., 2022) is a spectral-temporal graph neural network, one of the most competitive models without using SSL front-ends. The raw waveform is fed into convolution filters in the SincNet layer first. Then the following

RawNet2-based residual encoder produces a high-level representation. A max-pooling operation is applied to extract both temporal and spectral representations. The representations are fed into graph attention and graph pooling layers and merged into a heterogeneous spectral-temporal graph. The heterogeneous information is integrated into stack nodes in the heterogeneous stacking graph attention layers (HS-GALs). A max graph operation is applied to the outputs of HS-GAL to produce an advanced heterogeneous graph. Derived through the node-wise maximum and average operations over the graph, nodes are concatenated and fed into an FC layer to produce outputs.

Wav2Vec2-AASIST (Tak et al., 2022b) combines an informative SSL front-end and an AASIST classifier, achieving the SOTA performance. The front-end is a pre-trained wav2vec2 model XLS-R (0.3B) (Babu et al., 2022). The XLS-R model includes a convolutional encoder and a BERT-structured transformer. During pre-training, the discretized CNN encoder output is masked and fed into the transformer to generate the context output. A contrastive task for context output is conducted to distinguish the true discretized latent from distractors sampled from other time steps. The XLS-R model is pre-trained on 436K hours of bona-fide speech data in 128 languages. The back-end of Wav2Vec2-AASIST follows AASIST and consists of a RawNet2 encoder and graph-based modules.

## B. Implementation details

### B.1. HyperPotter architecture

All experiments are conducted on two NVIDIA H800 GPUs. We use the pretrained XLS-R 300M as our front-end. For FCM, the fuzzifier $m$ and the maximum iteration number are set to 2 and 5, respectively. The fusion balancing factors $\beta_1$, $\beta_2$ are set as 0.9 and 0.6.

All audio samples are either padded or truncated to 4 seconds (64,000 samples). Three data augmentation methods are applied: Rawboost (option 5: linear and non-linear convolutive noise and impulsive signal-dependent additive noise in series) (Tak et al., 2022a), SpecAugment (Park et al., 2019), and MUSAN (Snyder et al., 2015). We adopt the AdamW optimizer with a learning rate of 1e-6 and a weight decay of 1e-4. We employ a weighted cross-entropy loss to address the class imbalance in the ASVspoof 2019 LA training set, assigning a weight of 0.9 to the bona-fide class and 0.1 to the spoof class. We set the batch size to 96. The random seed is fixed to 1234. The model is trained for 100 epochs, and the checkpoint achieving the best performance on the In-The-Wild (ITW) set is selected.

Most hyperparameters follow default settings from Wav2Vec2-AASIST (4s input, 6 graph layers, lr, loss, epochs, node num $N$) and HGNN/ViHGNN (FCM fuzzifier $m$, max iterations $T$, hyperedge num $K \approx 0.3N$, and the degree-free setting). The remaining parameters are set either by hardware constraints (batch size) or by training loss trends (threshold $\tau$ and its warm-up schedule). The fusion factors ($\beta_1$ and $\beta_2$) were chosen during early model design and kept fixed thereafter.

### B.2. Checkpoint Selection

We used a randomly selected 2k subset of ITW only as a development set for checkpoint selection, because in our practice the EER on the ASV19LA dev set quickly drops below 0.1% (around epoch 8) and becomes uninformative for model selection. ITW dev set was not used for hyperparameter tuning and the reported hyperparameter configuration is not the best choice for minimizing ITW dev EER. For example, Table 3 shows that setting (D=0.25N) yields a better ITW EER (4.97%).

To improve fairness, we reran the model using two disjoint development sets (2k from PartialSpoof and 2k from ASV5), neither of which overlaps with any reported test set. As shown in Table 4, although absolute EER values increase, HyperPotter remains better than the baseline on most datasets, including ITW and FoR.

*Table 4.* Development-set replacement experiments, where ITW is replaced by PartialSpoof v1 or ASVspoof5 (2k randomly selected samples), reported in EER (%).

| Model | In-the-Wild | ASV20 19LA | ASV20 21LA | ASV20 21DF | ASV spoof5 | FoR | Codecfake | ADD22 Track1 | ADD22 Track3 | ADD23 R1 | ADD23 R2 | Libri Voc | SONAR |
|---|---|---|---|---|---|---|---|---|---|---|---|---|---|
| **Wav2Vec2+AASIST (ep80 | ITW dev)** | 7.58 | 0.26 | 2.48 | 4.08 | 13.38 | 4.24 | 40.22 | 33.79 | 16.14 | 25.55 | 22.21 | 6.96 | 32.02 |
| **HyperPotter (ep45 | ITW dev)** | **5.72**$_{\downarrow 1.86}$ | 0.23$_{\downarrow 0.03}$ | 3.94$_{\uparrow 1.46}$ | **1.78**$_{\downarrow 2.30}$ | 16.04$_{\uparrow 2.66}$ | **3.89**$_{\downarrow 0.35}$ | 34.47$_{\downarrow 5.75}$ | 32.34$^{\dagger}_{\downarrow 1.45}$ | **11.31**$_{\downarrow 4.83}$ | 21.49$_{\downarrow 4.06}$ | 21.84$^{\dagger}_{\downarrow 0.37}$ | 2.55$_{\downarrow 4.41}$ | 27.71$_{\downarrow 4.31}$ |
| **Wav2Vec2+AASIST (ep52 | PS dev)** | 8.79 | 0.27 | 2.25 | 4.62 | 12.94 | 4.68 | 40.93 | 34.60 | 15.07 | 25.02 | 21.13 | 8.58 | 33.03 |
| **HyperPotter (ep76 | PS dev)** | 6.67$_{\downarrow}$ | 0.31$_{\uparrow}$ | 3.58$_{\uparrow}$ | 2.00$_{\downarrow}$ | 17.38$_{\uparrow}$ | 4.64$_{\downarrow}$ | 36.32$_{\downarrow}$ | 33.70$_{\downarrow}$ | 11.97$_{\downarrow}$ | 20.85$_{\downarrow}$ | 20.66$_{\downarrow}$ | 3.82$_{\downarrow}$ | 26.04$_{\downarrow}$ |
| **Wav2Vec2+AASIST (ep52 | ASV5 dev)** | 9.26 | 0.34 | 2.79 | 4.48 | 12.77 | 5.96 | 41.12 | 35.26 | 15.45 | 24.84 | 21.67 | 9.90 | 36.75 |
| **HyperPotter (ep49 | ASV5 dev)** | 7.12$_{\downarrow}$ | 0.31$_{\downarrow}$ | 2.69$_{\downarrow}$ | 2.60$_{\downarrow}$ | 14.26$_{\uparrow}$ | 3.80$_{\downarrow}$ | 37.87$_{\downarrow}$ | 31.96$_{\downarrow}$ | 11.30$_{\downarrow}$ | 24.17$_{\downarrow}$ | 24.45$_{\uparrow}$ | 4.66$_{\downarrow}$ | 27.89$_{\downarrow}$ |

### B.3. HOI Distribution Visualization

Following Urbina-Rodriguez et al. (Urbina-Rodriguez et al., 2026), we conduct the HOI visualization experiment with three key design choices:

1. **Variables.** Each HOI unit is represented by a variable (typically scalar for efficiency). Here, we define the $k$-th variable as the cosine similarity between the prototype-aligned FCM centroid $\mathbf{c}_k$ and its aligned global prototype $\mathbf{p}_k^{(g)}$: $v_k = \cos(\mathbf{c}_k, \mathbf{p}_k^{(g)})$. This is motivated by: 1) centroid-level units are natural for probing hyperedge relations in our hypergraph model, and 2) centroid-prototype alignment degree is a permutation-invariance scalar and demonstrated effective in Figure 4b.

2. **Metrics.** For a triplet $S \subseteq \{1, \ldots, K\}$ with $|S| = 3$ and variables $\mathbf{v}_S = \{v_k\}_{k \in S}$, we compute (i) class-conditional O-information $\Omega(\mathbf{v}_S \mid y = c)$ to characterize within-class HOI structure, and (ii) label-relevant HOI via the O-information gain $\nabla_y \Omega(\mathbf{v}_S) = \Omega(\mathbf{v}_S \cup \{y\}) - \Omega(\mathbf{v}_S)$, where $y \in \{0, 1\}$ is the binary label. Triplets are used to capture genuine HOIs while keeping computation tractable.

3. **Normalization and visualization.** To aggregate unit-level statistics, we assign each triplet's O-information value to its participating variables and compute each unit's HOI score by averaging over all triplets that contain that unit. We then rank min–max normalize the unit scores for scale-invariant comparison across layers, and visualize them as layer-wise heatmaps in Figure 3.

## C. Datasets

Following a similar evaluation protocol to the Speech DF Arena leaderboard (Dowerah et al., 2026), we conduct experiments on 13 test sets, including ASVspoof 2019 LA (Todisco et al., 2019), In-the-Wild (Müller et al., 2022), ASVspoof 2021 LA, ASVspoof 2021 DF (Yamagishi et al., 2021), ASVspoof2024 (Wang et al., 2024), FoR (Reimao & Tzerpos, 2019), Codecfake (Xie et al., 2025), ADD2022 Track 1 and Track 3 (Yi et al., 2022), ADD2023 Track1.2 Round 1 and Round 2 (Yi et al., 2023), LibriVoc (Sun et al., 2023), and SONAR (Li et al.). The DFADD dataset is excluded due to a mismatch between its data format and our detection model.

*Table 5.* Summary of speech spoofing datasets used in this work.

| Dataset | Language | Source Dataset | Codec | Spoofing Methods |
|---|---|---|---|---|
| 19LA | English | VCTK | no | Traditional TTS/VC |
| 21LA | English | VCTK | Telephony/VoIP codecs | Traditional TTS/VC |
| 21DF | English | VCTK | Compressed audio | Diverse TTS/VC |
| LibriVoc | English | LibriTTS | no | Neural vocoders |
| ADD2022T1 | Chinese | AISHELL | no | Unspecified and complex |
| ADD2022T3 | Chinese | AISHELL | no | Unspecified and complex |
| ADD2023T1.2R1 | Chinese | AISHELL, THCHS-30 | no | Unspecified and complex |
| ADD2023T1.2R2 | Chinese | AISHELL, THCHS-30 | no | Unspecified and complex |
| ASVSpoof5 | English | MLS-English | Multiple codecs | complex TTS/VC |
| In-The-Wild | English | Public figures (online) | no | Online video segmentation |
| For | English | online resource | no | Commercial/public TTS |
| SONAR | Mixed | online resource | no | TTS incl. codec-based |
| CodecFake | Mixed | LibriTTS,VCTK,AISHELL | no | Neural codec models |

## D. Supplementary Evaluation results

### D.1. F1 score Comparison of Different Models

Table 6 reports F1-score results on the 13 evaluation sets as a supplementary comparison to the EER results in the main text. Baseline models for which F1-scores are not reported in the original papers are excluded from the table.

*Table 6.* F1-score (%) comparison on multiple evaluation sets. Higher is better. **Bold**, underline, and [†] indicate the best, second-best, and third-best results, respectively.

| Model | Params (M) | In-the-Wild | ASV20 19LA | ASV20 21LA | ASV20 21DF | ASV20 24 | FoR | Codecfake | ADD22 Track1 | ADD22 Track3 | ADD23 R1 | ADD23 R2 | Libri Voc | SONAR |
|---|---|---|---|---|---|---|---|---|---|---|---|---|---|---|
| Wav2Vec2+AASIST (Tak et al., 2022b) | 317.84 | 90.9 | 98.9 | 96.0 | 44.0 | 67.7 | 92.6 | 44.3 | 56.0 | 62.3 | 78.8 | 84.0 | 69.3 | 57.4 |
| XLSR+Conformer+TCM (Truong et al., 2024) | 319 | 93.7 | **99.1** | 86.6 | 71.8 | 63.7 | 89.3 | 52.0 | 49.0 | 55.3 | 82.4 | 83.4 | 92.2 | 76.1[†] |
| XLSR+BiCrossMamba (El Kheir et al., 2025) | 318.21 | 93.6 | 96.7 | 83.4 | 69.9 | 72.0 | 93.1 | 50.1 | **56.7** | 58.8 | 77.4 | 77.6 | 92.9 | 75.3 |
| XLSR+SLS (Zhang et al., 2024a) | 340 | 94.0[†] | 98.9 | 87.1 | 74.1[†] | 63.8 | 94.9[†] | **54.8** | 52.8 | 63.7 | **85.6** | 84.7 | **93.4** | 77.8 |
| XLSR+Mamba (Xiao & Das, 2025) | 319 | 94.6 | 98.0 | 95.5 | 74.4 | 70.8[†] | 93.3 | 52.8[†] | 52.4 | 57.7 | 83.6[†] | **85.4** | 92.6[†] | **78.2** |
| **Wav2Vec2+AASIST (Baseline)** | 317.84 | 93.9 | 98.8[†] | 88.7[†] | 56.7 | **72.5** | 95.8 | 47.5 | 52.9 | 63.0[†] | 80.6 | 83.8 | 79.3 | 71.0 |
| **HyperPotter** | 317.87 | **95.4** ↑1.5 | 98.9 ↑0.1 | 83.0 ↓5.7 | **75.4** ↑18.7 | 68.1 ↓4.4 | **96.1** ↑0.3 | 53.7 ↑6.2 | 54.6[†] ↑1.7 | **72.0** ↑9.0 | 83.9 ↑3.3 | 84.1[†] ↑0.3 | 91.6 ↑12.3 | 75.0 ↑4.0 |

## D.2. Varied speech patterns

We also evaluate HyperPotter on more varied speech patterns (EmoFake, MLAAD, SingFake) and report improvements over the baselines in Table 7. Accuracy is additionally reported for the fake-only dataset (MLAAD).

*Table 7.* Evaluation on additional diverse datasets, reported in EER (%) and Accuracy (%). All models are trained with ASV19LA.

| Model | Wav2Vec2+AASIST (Baseline) | HyperPotter | Wav2Vec2+VIB (Doan et al., 2024) |
|---|---|---|---|
| **EmoFake** *EER (%)* | 2.01 | **1.4** | - |
| **MLAAD(+MAILABS)** *EER (%)* | 23.16 | **18.23** | - |
| **SingFake** *EER (%)* | 45.83 | **40.77** | - |
| **MLAAD** *ACC (%)* | 76.47 | **81.68** | 72.44 |

## D.3. Generalization performance across diverse scenarios

Table 8 compares the F1-scores of the baseline model (Pairwise) and the proposed HyperPotter (HOI) across different scenarios. HyperPotter consistently outperforms the baseline in most scenarios, particularly in "Various Attacks", "Cross-Lingual", and "Varied Speakers". However, in the "Strong Distortion" scenario, HyperPotter shows a decrease in performance compared to the baseline. These results are visualized in Figure 1, which plots the F1-scores of both models across all scenarios, with the coordinate range set from 50% to 100%, and the visualization is shown on the right side of the figure.

*Table 8.* Comparison of F1-scores (%) between Pairwise (Baseline) and High-order (HyperPotter) interactions across different scenarios. Each scenario contains multiple datasets, and the performance is reported as the mean F1 score across these datasets.

| Scenario | Dataset(s) | Pairwise (Baseline) | HOI (HyperPotter) |
|---|---|---|---|
| Online Data | ITW, FoR, SONAR | 86.9 | 88.8 |
| Various Attacks | ADD22T1, ADD22T3, ADD23R1, ADD23R2, 21DF | 67.4 | 74.0 |
| Cross-Lingual | ADD22T1, ADD22T3, ADD23R1, ADD23R2, CodecFake, SONAR | 64.7 | 70.6 |
| Strong Distortion | 21LA, ASV2024 | 80.1 | 75.6 |
| Varied Speakers | LibriVoc | 79.3 | 91.6 |

## D.4. Comparison of Convergence Speed across Different Detection Models

Figure 5 compares the convergence behavior of different detection models. Although introducing HAGNN layers increases optimization difficulty, prototype-guided initialization stabilizes hyperedge construction and leads to a smoother loss decrease, narrowing the convergence gap between HyperPotter and the pairwise baseline.

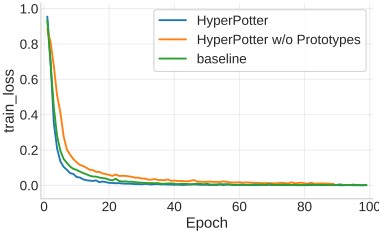

*Figure 5.* Comparison of loss curves across different detection models.

## D.5. Complete Visualization Results across Different Layers

As a supplement to Figure 4, we provide complete layer-wise visualization results to further illustrate the learned high-order relational patterns across datasets in Figure 6 and Figure 7.

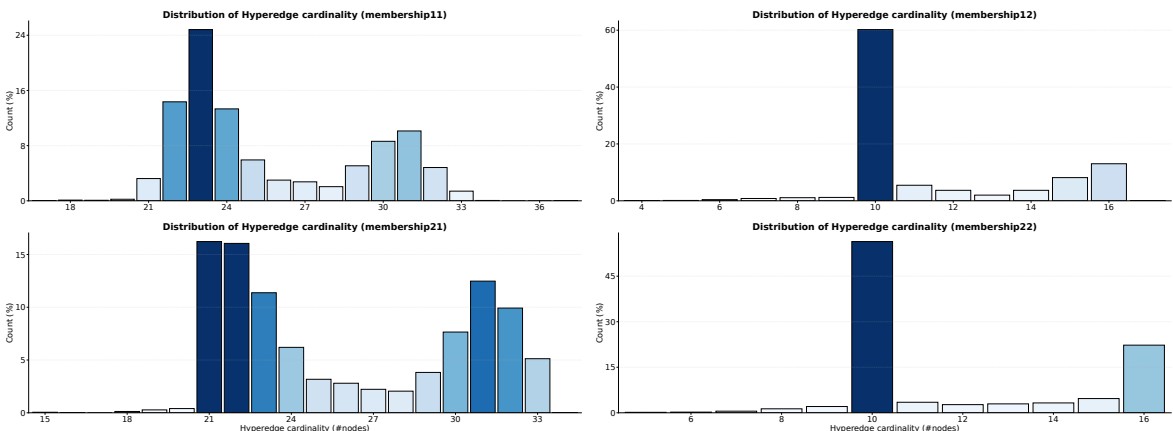

*Figure 6.* Distributions of hyperedge cardinality for four heterogeneous graph layers.

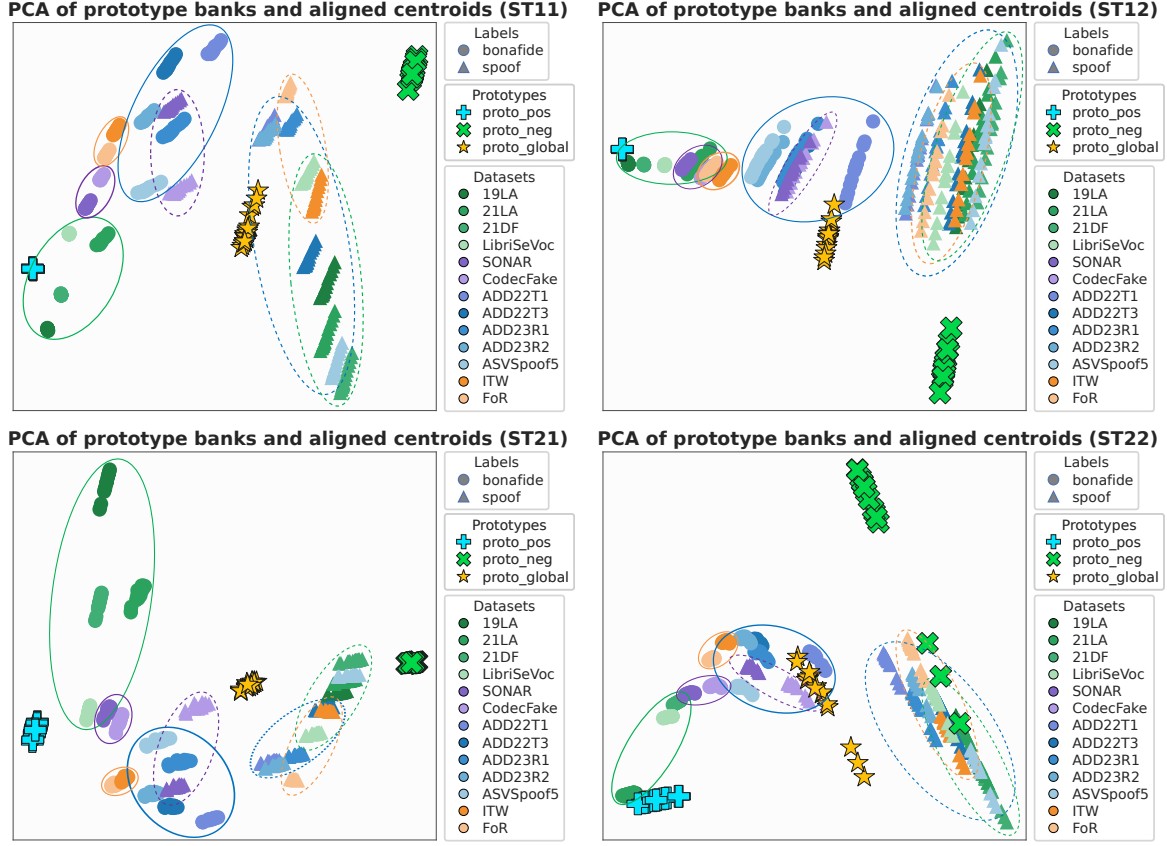

*Figure 7.* PCA of prototype banks and aligned centroids for four heterogeneous graph layers.

# E. Ablations

## E.1. Complete Ablation Results

Table 9 provides the complete ablation results across all 13 evaluation sets.

*Table 9.* Ablation study on HyperPotter sub-modules and alignment algorithms, reported as EERs (%).

| Model | In-the-Wild | ASV20 19LA | ASV20 21LA | ASV20 21DF | ASV spoof5 | FoR | Codecfake | ADD22 Track1 | ADD22 Track3 | ADD23 R1 | ADD23 R2 | Libri Voc | SONAR |
|---|---|---|---|---|---|---|---|---|---|---|---|---|---|
| **Wav2Vec2+AASIST(Baseline)** | 7.58 | 0.26 | 2.48 | 4.08 | **13.38** | 4.24 | 40.22 | 33.79 | 16.14 | 25.55 | 22.21 | 6.96 | 32.02 |
| **HyperPotter** | 5.72 | 0.23 | 3.94 | 1.78 | 16.04 | 3.89 | 34.47 | 32.34 | 11.31 | 21.49 | 21.84 | 2.55 | 27.71 |
| w/o Amplification Module (M2) | 6.90 ↑ | 0.23 | 3.50 ↓ | 1.88 ↑ | 14.46 ↓ | 6.45 ↑ | 36.45 ↑ | 33.52 ↑ | 15.67 ↑ | 21.03 ↓ | 22.36 ↑ | 3.43 ↑ | 32.98 ↑ |
| w/o A transposition (w/ A) | 5.25 ↓ | 0.33 ↑ | 3.66 ↓ | 2.28 ↑ | 16.78 ↑ | 4.24 ↑ | 37.47 ↑ | 33.27 ↑ | 13.82 ↑ | 23.85 ↑ | 24.36 ↑ | 3.43 ↑ | 27.36 ↓ |
| w/o $A^\top$ row-softmax (w/ col) | 6.09 ↑ | 0.20 ↓ | 3.83 ↓ | 1.73 ↓ | 18.19 ↑ | 3.62 ↓ | 35.32 ↑ | 34.63 ↑ | 14.53 ↑ | 23.04 ↑ | 25.68 ↑ | 3.15 ↑ | 34.17 ↑ |
| w/o Prototype Module (M3) | 6.49 ↑ | 0.26 ↑ | 2.80 ↓ | 1.88 ↑ | 15.42 ↓ | 4.59 ↑ | 36.73 ↑ | 32.87 ↑ | 11.59 ↑ | 22.10 ↑ | 23.87 ↑ | 3.86 ↑ | 29.15 ↑ |
| w/o Both Alignment Algs | 5.95 ↑ | 0.19 ↓ | 3.83 ↓ | 2.25 ↑ | 19.07 ↑ | 4.46 ↑ | 36.41 ↑ | 34.39 ↑ | 13.19 ↑ | 23.60 ↑ | 23.39 ↑ | 2.86 ↑ | 30.12 ↑ |
| w/o Soft Alignment Alg | 7.45 ↑ | 0.23 | 3.25 ↓ | 2.05 ↑ | 19.42 ↑ | 4.02 ↑ | 37.02 ↑ | 33.28 ↑ | 14.03 ↑ | 24.35 ↑ | 28.64 ↑ | 3.81 ↑ | 30.24 ↑ |
| w/o Slot Alignment Alg | 6.18 ↑ | 0.14 ↓ | 3.24 ↓ | 2.07 ↑ | 19.90 ↑ | 4.02 ↑ | 35.11 ↑ | 35.32 ↑ | 13.95 ↑ | 22.31 ↑ | 24.10 ↑ | 2.61 ↑ | 31.00 ↑ |

## E.2. Ablation Study for SSL-based front-end

Table 10 reports the ablation results obtained by replacing the HtrgGraphAttentionLayer in AASIST with the proposed HAGNN layer. To ensure fair comparison, all ablation experiments strictly follow the same training strategy and hyperparameter settings as used in AASIST, without any task-specific tuning for the newly introduced modules. Despite this, the proposed components consistently improve performance on several challenging benchmarks, including 21LA, 21DF, ASVspoof5, and ITW, demonstrating the effectiveness of hypergraph modeling and the prototype-based memory mechanism. Meanwhile, performance degradation is observed on certain in-domain datasets (e.g., 19LA), suggesting that further optimization of hyperparameters may lead to more consistent gains.

*Table 10.* Ablation results of replacing the HtrgGraphAttentionLayer in AASIST with HGNN and HGNN Memory (EER, %)

| Model | ITW | 19LA | 21LA | 21DF | ASVspoof5 |
|---|---|---|---|---|---|
| AASIST | 43.01 | **0.83** | 11.46 | 21.07 | 35.53 |
| AASIST + HGNN | 43.52 ↑ | 2.48 ↑ | **8.13** ↓ | 20.01 ↓ | **35.01** ↓ |
| AASIST + HGNN Memory | **38.65** ↓ | 3.10 ↑ | 10.66 ↓ | **18.64** ↓ | 35.30 ↓ |

# F. More Tests

## F.1. Impact of Codec Augmentations.

To enhance HyperPotter under severe channel interference, multiple codec strategies are investigated. The results are reported in Table 11. The conflicting performance trends between ITW and ASVspoof5 indicate codec-based methods improve performance in matched noisy conditions, while potentially interfering with HOI modeling and its generalization.

*Table 11.* Comparison of HyperPotter augmented with different codec strategies, reported as EERs (%).

| | ITW | 21LA | ASV5 | ADD22T1 | Codecfake |
|---|---|---|---|---|---|
| **HyperPotter** | **5.72** | 3.94 | 16.04 | 32.34 | 34.47 |
| **+speex** | 7.77 ↑ | **1.61** ↓ | 10.62 ↓ | 31.94 ↓ | 38.84 ↑ |
| **+opus** | 11.63 ↑ | 2.90 ↓ | **10.11** ↓ | 30.29 ↓ | 37.37 ↑ |
| **+Encodec** | 9.08 ↑ | 2.19 ↓ | 12.03 ↓ | 31.71 ↓ | 39.42 ↑ |

## F.2. Computational Complexity and Cost

To clearly present computational cost, below we summarize both time and memory complexity and empirical costs (latency/throughput/VRAM).

**Time complexity.** Let $N$ be the number of nodes, $F$ the feature dimension, $K$ the number of hyperedges, and $T$ the number of FCM iterations. The time complexity mainly consists of three parts:

- **Prototype-guided FCM.** Each iteration computes the distances between $N$ node features and $K$ centroids, costing $O(NFK)$ per iteration. Over $T$ iterations, this is $O(TNFK)$.

- **Amplification operator construction.** The structural term $A^{(c)} = UU^\top$ costs $O(N^2K)$. The feature term $A^{(f)} = X'X'^\top/\sqrt{F}$ costs $O(N^2F)$. Hence this step costs $O\left(N^2(K+F)\right)$.

- **Relational evidence propagation.** $Z = AX'$ and subsequent projections multiply the $N \times N$ operator $A$ with node features, which costs $O(N^2F)$ per propagation step.

Overall, the time complexity can be summarized as: $O\left(TNFK + N^2(K+F)\right)$.

**Memory complexity.** The memory complexity is mainly governed by the tensors in the three stages above:

- Storing $X'$: $O(NF)$, storing prototypes: $O(KF)$, storing memberships $U$: $O(NK)$.

- Constructing $A^{(c)}, A^{(f)}, A \in \mathbb{R}^{N \times N}$: $O(N^2)$.

- Storing propagated evidence $Z$: $O(NF)$.

Overall, the memory complexity can be summarized as: $O\left(N^2 + NK + NF + KF\right)$.

**Practical implications.** The computational complexity is primarily driven by $N$ via the $N^2$ components. In our setting, $N$ corresponds to the pooled node set produced by the graph builder (not raw frame-level tokens), so $N$ remains moderate in practice. $K$ is an effective efficiency knob: it impacts both the FCM cost $O(TNFK)$ and the structural operator cost $O(N^2K)$.

**Empirical validation.** We empirically validate these trends with controlled sweeps over $N/K/T$ in Table 12. Among the tested factors, reducing $K$ provides the most favorable efficiency–accuracy trade-off. Decreasing $K$ from the default to $0.25N$ reduces latency ($73.06 \rightarrow 60.27$ ms), improves throughput ($25.81 \rightarrow 38.52$ utt/s), and lowers peak memory ($14.63 \rightarrow 13.76$ GiB), while the mean EER in Table 3 changes only from about 6.27% to 6.74%. We also report overall inference/training latency, throughput, and peak VRAM for both baseline and our method in Table 13. Under the default setting, per-sample latency is well below real-time (73.06 ms), and the peak VRAM overhead is modest (14.63 GB).

*Table 12.* Computational-cost analysis of HyperPotter under three controlled sweeps on a server equipped with an AMD EPYC 7543 32-Core Processor and a single NVIDIA RTX A6000 GPU. **Default configuration**: cluster setting $(16, 16, 20, 16, 20, 16)$, 4 s input, and `max_iter` $= 5$. **The default setting in each sweep is highlighted in bold for easy comparison.** Latency is reported as per-sample latency with batch size $= 1$; throughput and peak memory are reported with batch size $= 128$. For the cluster sweep, the six values in each configuration correspond to $(S, T, \mathrm{ST11}, \mathrm{ST12}, \mathrm{ST21}, \mathrm{ST22})$.

| Experiment | Setting | Latency (ms) ↓ | Throughput (utt/s) ↑ | Peak Mem. (GiB) ↓ | Mean Iter. | Mean Nodes |
|---|---|---|---|---|---|---|
| | **Cluster-count sweep** (4 s input, `max_iter` $= 5$) | | | | | |
| $K$ | $0.25N$ $(11, 12, 14, 7, 14, 7)$ | 60.27 | 38.52 | 13.76 | 5.0 | 42.8 |
| $K$ | $0.50N$ $(21, 33, 27, 13, 27, 13)$ | 76.98 | 23.84 | 12.20 | 5.0 | 42.8 |
| $K$ | $0.75N$ $(32, 50, 41, 20, 41, 20)$ | 94.65 | 16.67 | 13.41 | 5.0 | 42.8 |
| $K$ | **HyperPotter** $(16, 16, 20, 16, 20, 16)$ | **73.06** | **25.81** | **14.63** | **5.0** | **42.8** |
| | **Node-count sweep** (default cluster setting, `max_iter` $= 5$) | | | | | |
| $N$ | 1 s input | 61.62 | 34.40 | 4.36 | 4.10 | 23.0 |
| $N$ | 2 s input | 62.03 | 34.99 | 7.49 | 5.00 | 29.5 |
| $N$ | **4 s input** | **69.34** | **27.20** | **14.97** | **5.00** | **42.8** |
| $N$ | 6 s input | 74.85 | 23.75 | 22.46 | 5.00 | 55.7 |
| $N$ | 8 s input | 81.37 | 20.97 | 29.94 | 5.00 | 69.5 |
| | **FCM iteration sweep** (4 s input, default cluster setting) | | | | | |
| $T$ | `max_iter` $= 1$ | 58.51 | 27.48 | 13.76 | 1.0 | 42.8 |
| $T$ | `max_iter` $= 3$ | 66.73 | 27.43 | 12.20 | 3.0 | 42.8 |
| $T$ | **`max_iter` $= 5$** | **69.30** | **27.34** | **13.41** | **5.0** | **42.8** |
| $T$ | `max_iter` $= 7$ | 76.53 | 27.16 | 14.63 | 7.0 | 42.8 |
| $T$ | `max_iter` $= 10$ | 86.32 | 26.79 | 15.85 | 10.0 | 42.8 |

*Table 13.* Inference and training cost comparison between the Wav2Vec2-AASIST(Baseline) and HyperPotter on a server equipped with an AMD EPYC 7543 32-Core Processor and a single NVIDIA RTX A6000 GPU.

| Batch size | Model | Inference | | | Training | | |
|---|---|---|---|---|---|---|---|
| | | Latency (ms) $\downarrow$ | Throughput $\uparrow$ | VRAM (MB) $\downarrow$ | Iter. time (ms) $\downarrow$ | Samples/s $\uparrow$ | Iter./s $\uparrow$ |
| 1 | Baseline | 19.49 | 51.35 | 2630.64 | 101.33 | 9.87 | 9.87 |
| | Ours | 77.50 | 12.90 | 2631.64 | 264.31 | 3.78 | 3.78 |
| 8 | Baseline | 85.97 | 93.05 | 4718.64 | 289.67 | 27.62 | 3.45 |
| | Ours | 348.67 | 22.94 | 4720.64 | 1199.85 | 6.67 | 0.83 |
| 16 | Baseline | 162.23 | 98.36 | 6421.08 | 511.28 | 31.29 | 1.96 |
| | Ours | 661.64 | 24.12 | 6422.08 | 2240.85 | 7.14 | 0.45 |
| 64 | Baseline | 606.74 | 105.20 | 12922.32 | 1768.53 | 36.09 | 0.57 |
| | Ours | 2523.79 | 25.29 | 13352.32 | 8563.60 | 7.45 | 0.12 |

## F.3. Stability

We assess stability of HyperPotter framework in two complementary senses:

1. **Construction-level stability under variations of key hypergraph parameters.** We report sensitivity to the hypergraph parameters $K$ (hyperedge count) and $\mathcal{R}$ (cardinality-related setting) in Table 3 and the maximum number of FCM iterations $T$ in Table 14. This indicates that HyperPotter's hyperedge construction is not overly sensitive to its primary hyperparameters, supporting construction-level stability.

*Table 14.* Sensitivity of model performance to maximum number of FCM iterations $T$, evaluated using EER (%).

| | ITW | 21LA | 21DF | ASVspoof5 | FoR |
|---|---|---|---|---|---|
| **T=5 (default)** | 5.72 | 3.94 | 1.78 | 16.04 | 3.89 |
| $T = 3$ | 5.79 | 3.19 | 2.25 | 19.95 | 5.21 |
| $T = 7$ | 5.87 | 3.97 | 2.20 | 20.06 | 4.59 |

2. **Training-level stability under different random seeds.** We evaluate HyperPotter under three different random seeds to quantify robustness to stochastic initialization and optimization. The results are listed in Table 15. Across random seeds, HyperPotter shows stable performance and preserves the overall trend of strong gains on most representative settings, with remaining limitations under codec-heavy conditions.

*Table 15.* Sensitivity of model performance to random initialization seeds across datasets, reported in EER (%). Results are reported under three random seeds for both the baseline and HyperPotter, and the mean $\pm$ std over seeds is shown in the last block. Lower EER indicates better performance, and the default seed is highlighted in bold.

| Seed | Model | ITW | 21LA | 21DF | ASVspoof5 | FoR |
|---|---|---|---|---|---|---|
| **1234 (default)** | Ours | 5.72 | 3.94 | 1.78 | 16.04 | 3.89 |
| | Baseline | 7.58 | 2.48 | 4.08 | 13.38 | 4.24 |
| 1111 | Ours | 6.54 | 3.06 | 2.35 | 17.27 | 5.96 |
| | Baseline | 8.43 | 1.82 | 4.28 | 13.57 | 12.10 |
| 4321 | Ours | 6.40 | 3.79 | 1.91 | 14.92 | 4.13 |
| | Baseline | 8.08 | 3.10 | 3.75 | 14.18 | 13.16 |
| Mean $\pm$ Std | Ours | 6.22 $\pm$ 0.44 | 3.60 $\pm$ 0.47 | 2.01 $\pm$ 0.30 | 16.08 $\pm$ 1.18 | 4.66 $\pm$ 1.13 |
| | Baseline | 8.03 $\pm$ 0.43 | 2.47 $\pm$ 0.64 | 4.04 $\pm$ 0.27 | 13.71 $\pm$ 0.42 | 9.83 $\pm$ 4.87 |

## F.4. Scalability

To evaluate the scalability of HyperPotter, we add longer-audio results (8s) in Table 16. HyperPotter remains competitive on most 8-second evaluation sets and improves over the 8-second baseline on several codec- or distortion-related scenarios, although degradation remains on some ADD tracks.

*Table 16.* Scalability experiments with longer audio inputs (8 s), reported in EER (%).

| Model | In-the-Wild | ASV20 19LA | ASV20 21LA | ASV20 21DF | ASV spoof5 | FoR | Codecfake | ADD22 Track1 | ADD22 Track3 | ADD23 R1 | ADD23 R2 | Libri Voc | SONAR |
|---|---|---|---|---|---|---|---|---|---|---|---|---|---|
| **Wav2Vec2+AASIST (4s \| ASV19)** | 7.58 | 0.26 | 2.48 | 4.08 | 13.38 | 4.24 | 40.22 | 33.79 | 16.14 | 25.55 | 22.21 | 6.96 | 32.02 |
| **HyperPotter (4s \| ASV19)** | **5.72**↓ | 0.23↓ | 3.94↑ | **1.78**↓ | 16.04↑ | **3.89**↓ | 34.47↓ | 32.34†↓ | **11.31**↓ | 21.49↓ | 21.84†↓ | 2.55↓ | 27.71↓ |
| **Wav2Vec2+AASIST (8s \| ASV19)** | 8.22 | 0.41 | 4.10 | 2.66 | 19.52 | 4.55 | 38.61 | 37.30 | 11.94 | 17.93 | 16.54 | 5.32 | 39.72 |
| **HyperPotter (8s \| ASV19)** | 5.81↓ | 0.17↓ | 2.04↓ | 2.15↓ | 14.63↓ | 3.40↓ | 35.89↓ | 32.45↓ | 12.76↑ | 20.10↑ | 21.16↑ | 2.81↓ | 35.50↓ |

## F.5. Visualization for sample-level relations.

To investigate the relational segments at the sample-level, we conduct experiments using the PartialSpoof dataset (Zhang et al., 2022). Specifically, we filter the dataset for utterances exceeding 5 s in duration that contain both bona-fide and spoofed segments. The audio waveforms are segmented into 0.64 s windows, for each of which a gap score is computed. Specifically, we extract HAGNN centroids from test segments and compute their cosine similarities to positive and negative prototypes for bona-fide and spoof classes. Based on these similarities, we define a distribution gap metric as $gap = (s_{bp} + s_{sn} - s_{bn} - s_{sp})/2$, which measures the relative discrepancy between class prototypes and segment representations. To assess inter-segment similarities, we generate clustermaps (Figure 8) to illustrate correlations via some representative samples, where deeper yellow intensity denotes higher similarity. The clustermap is accompanied by two color-coded sidebars: the outer bar represents the ground truth, while the inner bar indicates the predicted gap scores (blue for bona fide, red for spoof). For each utterance, we further apply K-Means clustering to the gap scores of all segments, where $k = 2$. The mean value of gap scores from the two cluster centroids is used as the classification threshold to distinguish bona-fide and spoofed segments. The corresponding classification results for the representative samples are illustrated in Figure 9.

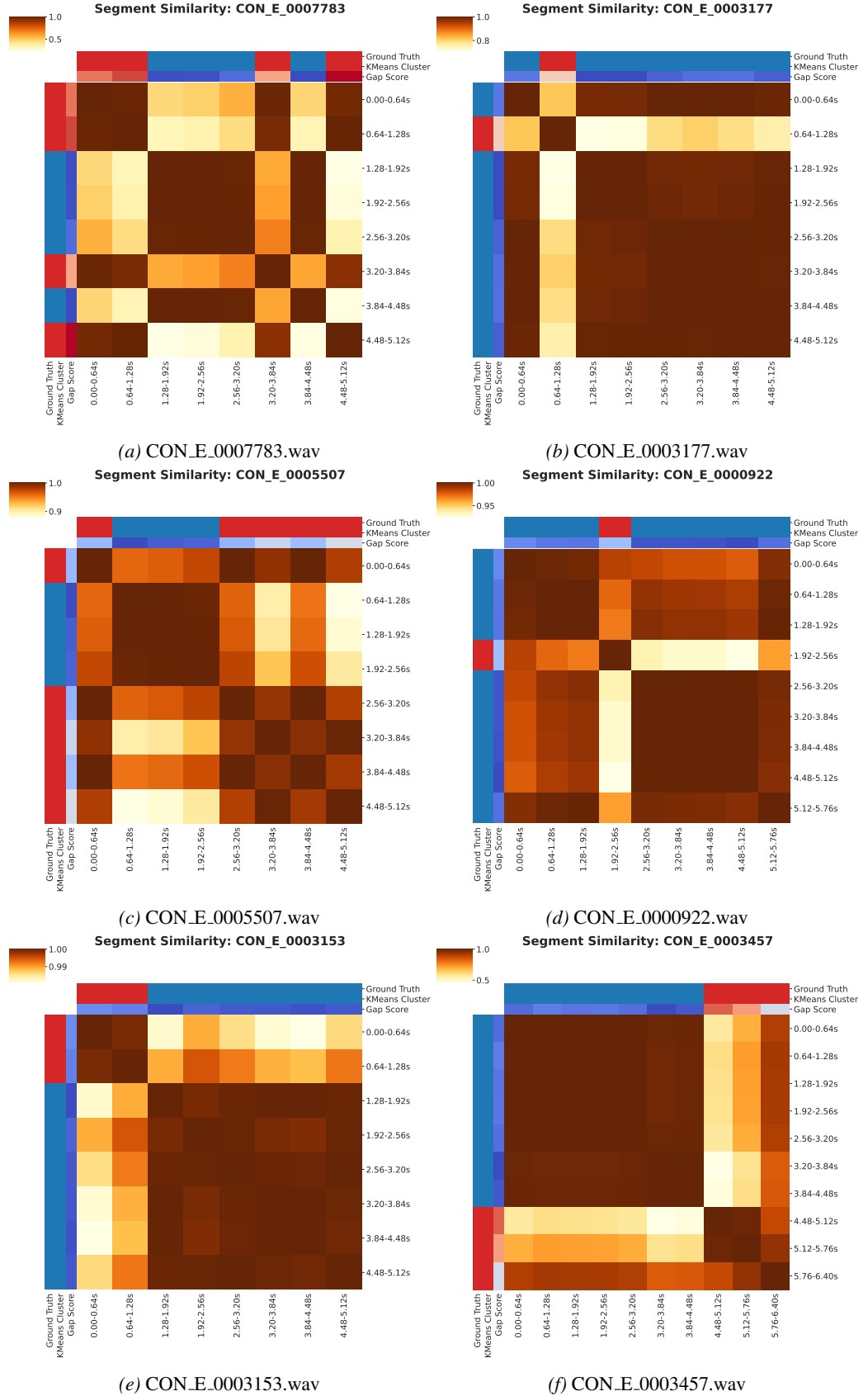

*(a)* CON_E_0007783.wav

*(b)* CON_E_0003177.wav

*(c)* CON_E_0005507.wav

*(d)* CON_E_0000922.wav

*(e)* CON_E_0003153.wav

*(f)* CON_E_0003457.wav

*Figure 8.* Clustermaps for different samples in PartialSpoof evaluation set

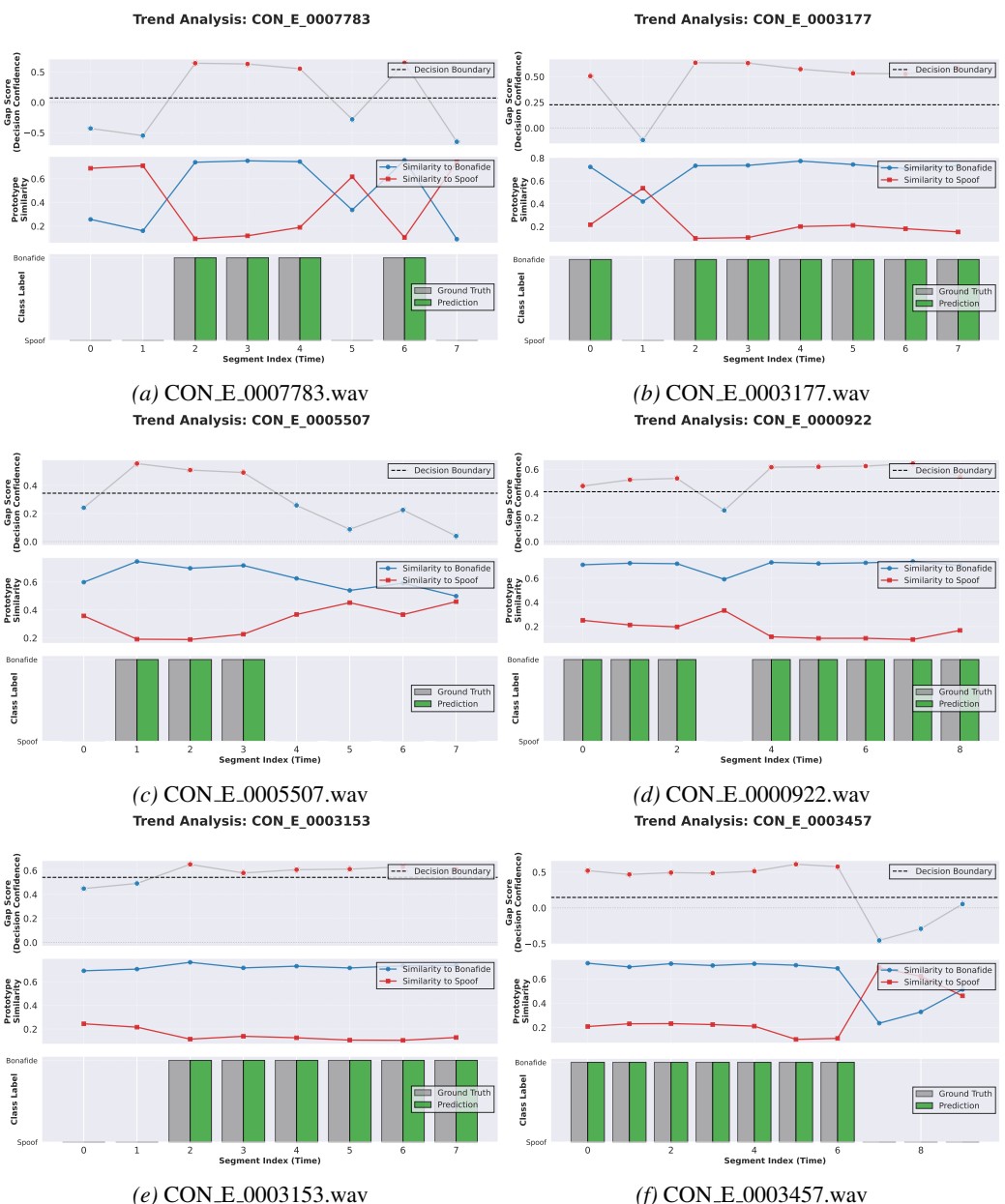

*Figure 9.* Classification results based on the distribution of centroids for different samples in PartialSpoof evaluation set

