# OpenReview forum: "HyperPotter: Spell the Charm of High-Order Interactions in Audio Deepfake Detection"
_ICML.cc/2026/Conference — ICML 2026 regular_

### Official Review · Reviewer_PWsk · 2026-03-05

**Soundness:** 3
**Presentation:** 2
**Significance:** 2
**Originality:** 3
**Overall Recommendation:** 4
**Confidence:** 4

**Summary:**

The paper proposes HyperPotter, a hypergraph-based audio deepfake detection framework designed to model high-order interactions across learned embeddings, motivated by an information theoretic discussion of O-information and the distinction between redundancy dominated and synergy dominated multivariate dependencies. The authors argue that many ADD systems rely on local patterns or pairwise relations, which may miss discriminative artifacts that only emerge when multiple feature components are considered jointly. They hypothesize that more recent and diverse deepfake artifacts contain meaningful synergistic high-order structure that supports better cross domain generalization. HyperPotter starts from a  baseline, Wav2Vec2 AASIST, using XLS-R and RawNet2 based encoders to form spectral and temporal node sets, then applies hypergraph attention layers with memory. Edges are constructed per sample by Fuzzy C-Means clustering to obtain soft node to edge memberships and centroids. A relational artifact amplification module then fuses structural similarity derived from the membership matrix with feature similarity of node embeddings, and applies an attention mechanism to emphasize evidence. A second key component is prototype guided hyperedge initialization. The method maintains prototype banks for bonafide, DF, and global prototypes, and uses them to initialize FCM centroids, with a training time alignment and EMA for stabilization and to accelerate clustering across batches. All models are trained only on ASVspoof2019 LA training data and evaluated on 13 test sets aligned with the Speech DF Arena style protocol, reporting EER and F1. The reported results show consistent gains over the retrained Wav2Vec2 AASIST baseline on many cross-scenario datasets such as In the Wild, 21DF, FoR, and ADD2022 Track 3, with some degradations on heavily codec affected sets like 21LA and ASVspoof5. The system is not tested in emotive, singing voice nor multilingual scenario (there are some cross-lingual experiments Chinese and English) scenario.

**Compliance With Llm Reviewing Policy:**

Affirmed.

**Final Justification:**

The authors presented an extensive clarification to the issues raised (both mine and other reviewers). The newly performed tests show that the trend on the performance is kept. I would ask the authors to find the space in the main body of the paper for at least part of the new tables, the description of the experiments made, as well as the clarification about the equations.

**Key Questions For Authors:**

1. Did you use the In The Wild test set only for selecting the final checkpoint, or also for any hyperparameter tuning decisions? If you rerun with a held-out validation set not overlapping any reported test set, how do results change, especially on ITW and FoR.? This would strongly affect confidence in the generalization claims.
2. Along which axis is the softmax applied to form A, and why is its transposition used in the final update rather than A. Do you have an ablation for the transposed A versus A and for different normalization choices?
3. Have you attemtped to compute O-information or a proxy on subsets of embeddings, and if not, what prevents it? A small scale measurement could substantiate the synergy motivation and clarify whether the gains truly correspond to increased synergistic dependence.
4. Is it possible to show the results of HyperPotter on some of the datasets mentioned above, to check the generalizibity claims on data with varied prosodic properties?

**Limitations:**

It would be improved by explicitly acknowledging protocol limitations, reporting computational costs, clarifying robustness boundaries under codecs and channel effects.

**Strengths And Weaknesses:**

Soundness: The clustering equations and core FCM formulation are correct, and the proposed components are plausible and empirically supported by many per dataset improvements and ablations. However, the information theoretic motivation is not validated by measuring O-information on model variables, so claims about synergy dominance are not established clearly.

Presentation: The overall narrative is easy to follow at a high level, with helpful diagrams and scenario based tables. Still, some crucial mathematical and algorithmic details are skipped or the equations contradict the text. The most important cases are mentioned below.

Significance: Improving out of domain generalization in audio deepfake detection is one of the mots important and timely problem. Modeling multi-way interactions is a meaningful direction that could influence detector design standards, especially for ensemble settings. The observed failure under strong codec distortion also identifies a practically relevant boundary where further research is needed. I would like to push the generalization aspect even further testing varied speech patterns, including emotion, singing and multilingual (not only cross-lingual on a significantly limited set of languages) analysis would significantly enhance the contribution and align with current interests in the area.

Originality: The use of hypergraph modeling with prototype guided initialization for ADD is a novel combination in this domain, and the prototype-bank design addresses a real implementation barrier for clustering based hyperedge construction. The O-information framing is conceptually interesting, though it currently functions more as motivation than as a demonstrated analytical tool within the paper.


Adding some of the aspects mentioned in the Significance and Originality would improve the impact of the paper, in particular adding at least some f the varied speech patterns and clearly showing the impact of O-information on the results of the model.


Strengths:
* Clear architectural hypothesis that multi way interactions matter for ADD, with a concrete hypergraph instantiation rather than vague discussion.
* Prototype guided initialization addresses a real practical issue with clustering based hyperedge construction, namely instability and cost from random initialization each batch
* Scenario level aggregation in Table 6 helps interpret where the method helps and where it hurts, rather than only listing per dataset scores.
* Ablations cover key modules including amplification and prototype banks, showing that gains are not purely from one minor trick. That includes also sensitivity analysis on hyperedge count and structural parameter.

Weaknesses:
* The largest weakness is that the core idea, O-information motivation is not empirically used in the paper. The paper claims evidence for synergy dominance without computing O-information or related multivariate measures on embeddings. This disconnect reduces the described motivation and theoretical base with the obtained results.
* Using RawNet2 encoder might not be the best solution, yet the models used in Table 1 for comparison are a good selection. I would consider adding model from [Thien-Phuc Doan, Hung Dinh-Xuan, Taewon Ryu, Inho Kim, Woongjae Lee, Kihun Hong, and Souhwan Jung. 2024. Trident of Poseidon: A Generalized Approach for Detecting Deepfake Voices. In Proceedings of the 2024 on ACM SIGSAC Conference on Computer and Communications Security (CCS '24). Association for Computing Machinery, New York, NY, USA, 2222–2235. https://doi.org/10.1145/3658644.369031 ]
* As for the datasets, the authors evaluate their solution on multiple datasets (13 different sets, but as for the characteristics of the data, some are quite similar - 4 subsets of ASVspoof family, and 4 of ADD) but they do not test it under various characteristics of the speech, e.g. emotional DF, singing voice spoofs or multilingual environment. Adding some results for EmoFake, EmoSpoof-TTS, MLAAD, SpeechFake, and SingFake would be beneficial.
* Limited analysis of computational cost and latency. FCM clustering per sample per layer can be expensive, yet the paper does not report runtime, memory overhead, or how cost scales with N, K, and iteration count.
* I would also add here lack of justification of using transposed A in the final amplification. Since A is non-symmetric after softmax, the operation is not intuitive and I would ask for expansion both paragraphs in Sect. 4.4 to make it more obvious.
* Another thing that could be explained in more detail is the soft alignement, where the description says „We employ a greedy matching algorithm based on cosine similarity to find the optimal centroid-prototype mapping”, yet the presented equation is argmax over $\pi$, suggesting global knowledge.
*  There is another ambiguity  when the authors in the Notation mention „The hypergraph contains K hyperedges, where each hyperedge connects all nodes, resulting in a degree-free hypergraph with hyperedge degree D.”  Later experiments treat D as a structural parameter controlling high order constraints and even state that D = 2 reduces to a pairwise graph, so connecting all nodes might need clarification. Additionally, I would consider renaming parameters to avoid confusion between D and $\mathcal{D}$.
 * A minor weakness is underspecified softmax axes and aggregation operators in the amplification module. Without specifying whether softmax is row-wise, column-wise, or global, the operator A is ambiguous, and so are stability and interpretation. One has to note that the codebase has been provided, so it is possible to retrace it.

---

> ### Author Rebuttal · Authors · 2026-03-31
>
> Thank you for the detailed and technically grounded review. We made concrete efforts to address the key weakness: the lack of empirical verification of the O-information motivation. We also added new generalization results and implementation-level clarifications (see attachment: https://anonymous.4open.science/r/HyperPotter-6C052325/r.pdf).
>
> 1. **O-information not measured (W1/Q3)**
>
>     We now provide empirical measurements characterizing the HOI structure in the learned representations. Specifically, we compute O-information metrics on layer-wise hyperedge centroids and visualize the results as synergy-redundance rank heatmaps (Fig. 9, attachment). We further compare HyperPotter with an architecture-matched untrained model, suggesting that our model learned structured high-order dependence. Together with the pairwise reduction (D=2) and the module-ablation drops, these results provide suggestive evidence that the observed gains are consistent with increased learned synergy. This is in line with recent findings that high-synergy components are more performance-critical [1].
>
> 2. **ITW checkpoint selection (Q1)**
>
>     We clarify that ITW was not used for hyperparameter tuning or architectural decisions. We used a randomly selected 2k subset of ITW only as a dev set for checkpoint selection, because in our practice the EER on the ASV19LA dev set quickly drops below 0.1% (around epoch 8) and becomes uninformative for model selection.
>
>     We provide two pieces of evidence that the ITW dev set was not used for hyperparameter tuning. (i) Most hyperparameters follow default settings from Wav2Vec2-AASIST (4s input, 6 graph layers, lr, loss, epochs, node num N) and HGNN/ViHGNN (FCM fuzzifier m, max iterations T, hyperedge num $K\approx 0.3N$, and the degree-free setting). The remaining parameters are set either by hardware constraints (batch size) or by training loss trends (threshold $\tau$ and its warm-up schedule). The fusion factors ($\beta_1$ and $\beta_2$) were chosen during early model design and kept fixed thereafter. (ii) The reported hyperparameter configuration is not the best choice for minimizing ITW dev EER. For example, Tab.3 shows that setting (D=0.25N) yields a better ITW EER (4.97%). Such a configuration would have been prioritized when tuning specifically on the ITW dev.
>
>     Following your suggestion, we reran the model using two disjoint development sets (2k from PartialSpoof and 2k from ASV5), neither of which overlaps with any reported test set. As shown in Tab.11 (attachment), although absolute EER values increase, HyperPotter remains better than the baseline on most datasets, including ITW and FoR.
>
> 3. **More baselines / datasets (W2/W3/Q4)**
>
>     In Tab.9 (attachment), we add “Trident of Poseidon” as a new baseline with two versions (w/ and w/o HAR). We also evaluate on more varied speech patterns (EmoFake, MLAAD, SingFake) and report improvements over the baselines (Tab.13, attachment).
>
> 4. **Computational cost and latency (W4)**
>
>     Our complexity is $O(ND_{in}(K^2+TK)+N^2(K+D_{out}))$, indicating that runtime is primarily driven by N and K. We empirically validate this with controlled sweeps over N/K/T (Table 15, attachment). We also report overall inference/training latency, throughput, and peak VRAM for both baseline and our method in Tab.14 (attachment).
>
>     The results suggest reducing K from the default setting to $0.25N$ can increase throughput to 38.52utt/s (+49.2%), and reduce peak memory to 13.76 GiB (−5.9%) with only minor accuracy degradation. Under the default setting, per-sample latency is well below real-time (73.06 ms), and the peak VRAM overhead is modest (14.63 GB).
>
> 5. **Softmax axis, why $A^\top$, and ablations (W5/W8/Q2)**
>
>     (i) Axis: In our code, the operator $A$ is normalized with softmax along the “source” dimension (row-normalized), so each node distributes its weight over other nodes. (ii) Why transpose: We implement propagation as a two-step update—“aggregate then project back”—by multiplying with $A$ and then with $A^\top$. This is intended to mirror the node→relation→node flow used throughout the model. (iii) Ablations: Attachment Tab.10 compares $A^\top$ vs. $A$ and row- vs. col-norm. Using $A$ can improve ITW but hurts FoR, while col-norm degrades ITW.
>
> 6. **“Greedy matching” equation mismatch (W6)**
>
>     We apologize for the confusion caused by the incorrect equation. Our greedy matching repeatedly selects the best unused centroid–prototype pair and locks it until all slots are assigned. This is not a global argmax. We will revise the equation.
>
> 7. **Ambiguity of D (W7)**
>
>     We will fix the notation D, remove the imprecise sentence “each hyperedge connects all nodes”, and clarify the “effective participation” of a hyperedge can be controlled by thresholding memberships.
>
> 8. **RawNet2 encoder (W2)**
>
>     We will note stronger encoders can be swapped in and RawNet2 is used to remain consistent with baseline.
>
> [1] https://arxiv.org/abs/2601.06851

---

> > ### Author Rebuttal · Reviewer_PWsk · 2026-04-01
> >
> > I would like to thank the authors for their response and enhanced tests their made. The authors have responded to vast majority of my questions, additional tests have been performed, hence if the rigidity of the tests and the issues of not so clear equations will be sufficiently described in the text of the paper I believe the grade should be increased.

---

> > > ### Author Response · Authors · 2026-04-07
> > >
> > > Thank you for the valuable response, and we sincerely appreciate your willingness to raise your recommendation. We also appreciate your note that clearer documentation of the test rigor and equation details would further strengthen the work.
> > >
> > > In the revised manuscript, we will revise the presentation carefully and add the missing implementation-level details you highlighted, including:
> > >
> > > - completing the HOI test details (variable construction, metric selection, O-information score normalization, and plotting protocol);
> > > - providing fuller details for ablations, cost/latency measurement, and checkpoint-selection procedures;
> > > - correcting the incorrect equation for the greedy matching procedure;
> > > - clarifying the ambiguous symbol D;
> > > - improving the presentation around operator normalization and the use of the transposed operator.
> > >
> > > We sincerely appreciate the opportunity during the rebuttal process to discuss technical details and address presentation issues. We believe the revised version, incorporating your feedback, will be substantially clearer, more rigorous, and easier to reproduce.
> > >
> > > Thanks again for your careful review and helpful guidance!

---

### Official Review · Reviewer_XQW4 · 2026-03-09

**Soundness:** 3
**Presentation:** 3
**Significance:** 3
**Originality:** 4
**Overall Recommendation:** 4
**Confidence:** 4

**Summary:**

This paper proposes HyperPotter, a hypergraph-based framework for audio deepfake detection that aims to improve cross-domain generalization by modeling HOIs among audio features. The paper is motivated by an information-theoretic discussion based on O-information, arguing that synergistic interactions beyond pairwise dependencies may be important for capturing synthetic artifacts. Building on a Wav2Vec2-AASIST-style backbone, the method introduces two main components: (1) an FCM-based hypergraph attention module for constructing and propagating hyperedge-level relational information, and (2) a prototype-guided hyperedge initialization mechanism with class-aware and global prototype banks to provide more stable clustering and long-term memory. Experiments are conducted by training on ASVspoof2019 LA and evaluating on a broad set of out-of-domain benchmarks. The paper reports improved performance over its baseline on many datasets, especially in challenging cross-scenario settings, and includes ablations and visualizations intended to support the benefit of higher-order relational modeling.

**Compliance With Llm Reviewing Policy:**

Affirmed.

**Final Justification:**

Thank you for the follow-up and for clearly summarizing the planned revisions. I appreciate the authors’ willingness to incorporate these changes in the revised manuscript. I have no further comments, and I maintain my original weak-accept recommendation.

**Key Questions For Authors:**

1. The paper’s motivation relies on the claim that synthetic speech artifacts arise from complementary manipulations across temporal dynamics, spectral structures, and latent feature representations. However, I did not find sufficient evidence supporting this claim. Since this point is central to the proposed higher-order formulation, the paper would be strengthened by either empirical support (e.g., ablations/analysis) rather than an established fact.

2. The interpretation in Section 3.1 feels somewhat too strong. In particular, the text appears to treat $C$ as “redundancy” and $B$ as “synergy” rather directly, whereas these quantities more generally capture different aspects of multivariate dependence and are not equivalent to pure redundancy/synergy in a formal PID sense. Similarly, $\Omega(X^n) > 0$ does not by itself imply that higher-order dependencies can be fully reduced to pairwise relations without additional assumptions. I would encourage the authors to soften this discussion and present it as heuristic motivation rather than a formal implication.

3. It is not fully clear to me whether the proposed framework explicitly models higher-order interactions beyond using hyperedges as an intermediate construction. In particular, the key operators in Sec. 4.4 are based on $A^{(c)} = UU^\top$ and $A^{(f)} = X'X'^\top / \sqrt{D}$, both of which are node-node pairwise affinity matrices, and the final fused operator $A$ is also pairwise. This makes it difficult to assess whether the higher-order structure is truly preserved and exploited, or whether the method effectively reduces to pairwise message passing after the hyperedge construction step. I would encourage the authors to clarify this point and better explain what part of the model is responsible for retaining genuinely higher-order information.

4. The trends in ablation study are not fully consistent: for example, removing the amplification module or the prototype banks appears to improve performance on 21LA and ASVspoof5. This suggests that the proposed components may be helpful in some scenarios but not uniformly beneficial. I would encourage the authors to moderate claims such as “consistently improves” and to discuss more explicitly why performance degrades under distortion/codec-heavy settings.

5. There seems to be some inconsistency in the headline performance numbers reported in different parts of the paper. For example, the abstract mentions an average relative gain of 22.15% across 11 datasets, whereas the conclusion reports 15.32%, and the experimental section states that evaluation is conducted on 13 test sets. It would be helpful for the authors to clarify exactly how these summary numbers are computed (e.g., which datasets are included and whether the metric is relative EER reduction or something else) and to ensure consistency throughout the paper.

**Limitations:**

yes

**Strengths And Weaknesses:**

### Strengths:

1. The paper tackles an important and practically relevant problem: robust cross-domain audio deepfake detection.

2. The proposed method is reasonably novel in introducing a hypergraph-based formulation with prototype-guided hyperedge initialization.

3. The empirical results show promising gains over the baseline on multiple out-of-domain test sets.


### Weaknesses:

1. The core motivation is plausible but under-supported. The paper argues that synthetic speech artifacts arise from complementary manipulations across temporal dynamics, spectral structures, and latent representations, which in turn motivates explicit higher-order modeling. However, this claim is not sufficiently substantiated by empirical evidence or analysis. Since it plays an important role in justifying the proposed formulation, the paper would benefit from either stronger experimental support or a more cautious framing of this point as a hypothesis rather than an established fact.


2. Some of the information-theoretic interpretation appears overstated. In Section 3.1, the discussion seems to map $C$ rather directly to redundancy and $B$ to synergy, and to suggest that redundancy-dominated systems are more naturally decomposable into pairwise relations. As written, this reads more strongly than what is formally supported. A more careful distinction between rigorous properties of O-information-related quantities and the heuristic intuition they provide would improve the theoretical clarity of the paper.


3. It is not fully clear whether the method truly preserves and exploits higher-order interactions. Although the framework is introduced as hypergraph-based, some key operators are ultimately expressed through node-node pairwise affinity matrices (e.g., $UU^\top$ and $X'X'^\top$). This makes it difficult to tell whether the model is genuinely leveraging higher-order structure or whether the hypergraph serves mainly as an intermediate construction before reducing back to pairwise message passing. Clarifying where the higher-order information is retained in the actual computation would make the method more convincing.


4. The empirical support, while promising, is not entirely consistent or comprehensive. The ablation study is conducted only on a subset of the evaluation datasets, and some components do not appear uniformly beneficial across conditions, especially under distortion/codec-heavy settings.

---

> ### Author Rebuttal · Authors · 2026-03-31
>
> Thank you for the thoughtful feedback. We respond with detailed clarifications and new results  (see the attachment: https://anonymous.4open.science/r/HyperPotter-6C052325/r.pdf).
>
> 1. **“Synthetic artifacts arise from complementary manipulations” motivation (W1/Q1)**
>
>     We agree that our original wording was too strong and we will present this as a hypothesis rather than an established fact.
>
> 2. **Overstated interpretation in Sec. 3.1 (W2/Q2)**
>
>     We will soften Sec. 3.1 as follows: (i) Total correlation $C$ and binding information $B$ are two measures of interdependence. Redundancy tends to contribute more strongly to $C$, while synergy tends to contribute more strongly to $B$ [1]. (ii) $Ω>0$ indicates that a system is redundancy-dominated, meaning variables share "copies" of the same information; this is distinct from being fully reducible to pairwise interactions. In the revision, we will explicitly present $C$, $B$ and $\Omega$ as heuristic or diagnostic measures rather than formal implications.
>
> 3. **Whether HOI is truly preserved (W3)**
>
>     We now provide direct measurements on learned representations. We compute O-information–based metrics [2] on our hyperedge centroids and visualize layer-wise heatmaps to estimate synergy-redundance distribution of HOIs (Attachment Fig. 9). The visualization further compares HyperPotter with an architecture-matched untrained model (Fig. 9b), suggesting that our model learned structured high-order dependence.
>
> 4. **HOI Preservation in Operator $A$ (W3/Q3)**
>
>     We clarify that while the operator $A$ is expressed as a pairwise matrix, the high-order information is already encoded in its structural term $A^{(c)}=UU^\top$ and in the node features $X$ during the former hyperedge construction stage.
>
>     **Specifically**, the FCM-based hypergraph module aggregates multi-node dependencies to update the membership $U$ and the features $X$. The term $UU^\top$ is not a simple pairwise similarity, but captures shared hyperedge participation, reflecting whether nodes belong to the same high-order groups. The subsequent multiplication $AX$ (including $A^{(c)}X$) uses this structural evidence to reinforce node features.
>
>     **Overall**, the amplification step does not reduce the model to simple pairwise message passing, because both $U$ and $X$ are induced by multi-node hyperedge aggregation. We will add a short clarification paragraph in Sec. 4.4 to make this dependence explicit.
>
> 5. **Non-uniform gains under codec/distortion (W4/Q4)**
>
>     We agree and will moderate “consistently improves”–style phrasing. Empirically, heavy codec/channel interference can mask fine-grained higher-order structure; in such settings, redundant dependence can be more robust than emphasizing synergy. We will add this as an explicit robustness-boundary discussion and interpret the observed ablation trends accordingly, rather than implying universal gains.
>
> 6. **Ablations coverage (W4)**
>
>     We now report results across all 13 test sets and add new ablations for the amplification operator (normalization/transposition variants) in Table 10 (attachment) to address the incompleteness concern. These modules performs well on the newly added benchmarks, indicating their effectiveness on cross-lingual and neural-codec scenarios.
>
> 7. **Inconsistent headline numbers (Q5)**
>
>     We apologize for the confusing reporting and for the arithmetic error. “22.15% over 11” is the average relative EER reduction excluding the two test sets where performance drops (21LA and ASVspoof5). When we compute the same metric over all 13 evaluated test sets, the correct average relative EER reduction is 12.68% (the paper mistakenly reports 15.32% due to an oversight). We will unify the aggregation rule and ensure consistent reporting throughout the paper.
>
>
> We believe these changes make the theoretical discussion appropriately cautious while improving clarity and strengthening the empirical support.
>
> [1] Schneidman, Elad, et al. "Network information and connected correlations." Physical review letters 91.23 (2003): 238701.
>
> [2] https://pypi.org/project/hoi/

---

> > ### Author Rebuttal · Reviewer_XQW4 · 2026-04-02
> >
> > Thank you for the detailed rebuttal. The main concerns raised in my review have been addressed satisfactorily, especially regarding the softened theoretical framing, the clarification of higher-order interaction preservation, the broader ablation coverage, and the correction of the summary statistics.
> >
> > I will maintain my original score, since the rebuttal mainly strengthens my confidence in my original weak-accept assessment rather than materially changing it.

---

> > > ### Author Response · Authors · 2026-04-07
> > >
> > > Thank you for taking the time to read our rebuttal and for confirming that your concerns have been adequately addressed. We appreciate your suggestions on making the motivation and information-theoretic discussion more precise and appropriately cautious.
> > >
> > > In the revised manuscript, we will incorporate the following changes:
> > >
> > > - reframe our motivation explicitly as a hypothesis rather than an established claim;
> > > - include direct HOI measurements on learned hypergraph centroids;
> > > - clarify where and how higher-order structure is retained despite the pairwise-form operators;
> > > - expand ablations and more explicitly discuss the limitations under codec/distortion-heavy conditions;
> > > - correct and unify the summary statistics to remove inconsistencies.
> > >
> > > Thank you again for your highly valuable feedback! We believe it will help make the revision more clearly motivated and more empirically grounded.

---

### Official Review · Reviewer_RiiH · 2026-03-12

**Soundness:** 2
**Presentation:** 3
**Significance:** 2
**Originality:** 3
**Overall Recommendation:** 4
**Confidence:** 3

**Summary:**

This paper proposes HyperPotter, a hypergraph-based framework for audio deepfake detection that explicitly models high-order interactions (HOIs) among features. Motivated by O-information theory, the authors argue that synergistic relationships among multiple acoustic representations provide stronger signals for identifying synthetic speech. The proposed approach constructs prototype-guided hyperedges and introduces a module for amplifying relational artifacts. Experiments across multiple datasets demonstrate improvements in cross-domain detection performance.

**Compliance With Llm Reviewing Policy:**

Affirmed.

**Key Questions For Authors:**

1.	How sensitive is the framework to the hyperedge construction parameters?
2.	Can the authors clarify how O-information directly influences model learning?
3.	How does the method scale with larger datasets or longer audio sequences?
4.	Could the authors compare against additional recent transformer-based spoof detection models?

**Limitations:**

Yes

**Strengths And Weaknesses:**

Strengths
●	The paper addresses an important problem: robust detection of AI-generated audio.
●	Modeling high-order interactions via hypergraphs is an interesting direction.
●	The authors attempt to provide theoretical motivation using O-information theory.
●	Experiments cover multiple datasets and cross-domain scenarios.
●	The framework combines prototype learning and hypergraph modeling in a coherent architecture.

Weaknesses
1.	The methodological novelty appears limited since hypergraph learning and prototype initialization have been widely studied.
2.	The connection between O-information theory and the actual model optimization is not clearly formalized.
3.	Hyperedge construction details are insufficiently explained, making the method difficult to reproduce.
4.	Baseline comparisons omit several strong modern approaches (e.g., SSL-based speech representations).
5.	The ablation study is limited and does not isolate the impact of each module.
6.	Computational complexity and scalability analysis are missing.

---

> ### Author Rebuttal · Authors · 2026-03-31
>
> Thank you for the constructive review. We address your main concerns with detailed clarifications and new results (see the attachment: https://anonymous.4open.science/r/HyperPotter-6C052325/r.pdf).
>
> 1. **Novelty beyond “hypergraph + prototypes” (W1)**
>
>     Our contribution is not the use of hypergraphs or prototypes, but **a task-driven design for ADD that makes higher-order feature interactions explicit and stable**. In ADD, artifacts are subtle and distributed across time/frequency cues. Many prior systems combine cues via feature- or score-level fusion, which often depends on hand-crafted combination rules. However, such hand-crafted rules limit the modeling of diverse and dynamic feature interactions, motivating the adoption of a learnable structure that can explicitly learn complementary high-order interactions.
>
>     **We address this gap with:** (a) per-layer FCM hyperedges that capture higher-order grouping patterns dynamically, (b) an attention module that amplifies informative interactive signals, and (c) prototype banks that act as cross-batch structural priors to stabilize hyperedge construction. We will revise the paper to highlight this ADD-specific gap and clarify how explicit HOI modeling benefits the system.
>
> 2. **“How O-information directly influences model learning” (W2/Q2)**
>
>     O-information is not directly optimized as a training objective in our current model; instead, it shapes our approach as **(i) a design motivation** and **(ii) an empirical diagnostic**.
>
>     **First,** because O-info characterizes the balance between synergistic and redundant HOI, it motivates our hypothesis that ADD artifacts may involve HOI beyond pairwise relations, and thus supports adopting a hypergraph structure that can naturally encode HOIs (hypergraphs can promote HOI estimation in complex systems [1]).
>
>     **Second,** to make this connection empirical rather than purely motivational, we compute O-information metrics [2] on our hypergraph centroids and visualize them as layer-wise synergy-redundance rank heatmaps (Fig. 9, attachment). The visualization further compares HyperPotter with an architecture-matched untrained model (Fig. 9b), suggesting that our model learned structured high-order dependence.
>
>     Accordingly, we will revise the paper to state this scope explicitly: O-information motivates the hypergraph design and serves as a diagnostic to probe synergistic-redundant HOI distributions. We leave direct O-information–regularized optimization to future work.
>
> 3. **Hyperedge construction details / reproducibility (W3)**
>
>     We will expand Sec. 4.3 with a step-by-step description of hypergraph construction, focusing on the FCM clustering procedure and the node update rules. To improve reproducibility, we have released our code, and we will also add pseudocode.
>
> 4. **SSL baselines (W4/Q4)**
>
>     Following your suggestion, we will emphasize comparisons with strong modern transformer-based SSL backbones (e.g., XLSR/WavLM encoders) and add the recent “Trident of Poseidon” baseline in Table 9 (attachment).
>
> 5. **Ablations limited (W5)**
>
>     Our model contains three main components: the hypergraph module (M1), the artifact amplification module (M2), and the prototype module (M3). Table 2 already reports ablations on M2/M3 as well as the alignment sub-components within M3. We add new ablations for sub-components within M2 in Table 10 (attachment), including operator and normalization choices. Overall, these modules are beneficial on most benchmarks, and removing M2 causes the largest degradations on ITW and FOR.
>
> 6. **Impact of hyperedge parameters (Q1)**
>
>     Consistent with ViHGNN, we evaluate sensitivity to hyperedge count and degree (Table 3 in the paper). The small EER fluctuations indicate that our model is relatively insensitive to these hyperedge parameters.
>
> 7. **Scalability and computational complexity (W6 / Q3)**
>
>     We add longer-audio results (8s) in Table 12 (attachment). HyperPotter remains competitive across most datasets, including distortion scenarios.
>
>     We also report overall inference/training latency, throughput, and peak VRAM for both baseline and our method in Table 14 (attachment). Under the default setting, per-sample latency is well below real-time (73.06 ms), and the peak VRAM overhead is modest (14.63 GB).
>
>
> We hope these revisions address your concerns on both significance and soundness, while substantially improving clarity and reproducibility.
>
> [1] Battiston, Federico, et al. “Networks beyond pairwise interactions: Structure and dynamics.” *Physics Reports* 874 (2020): 1–92.
>
> [2] https://pypi.org/project/hoi/

---

> > ### Author Rebuttal · Reviewer_RiiH · 2026-04-05
> >
> > A partially satisfied rebuttal is provided by the authors, mainly establishing the stability & computational complexity are not resolved.

---

> > > ### Author Response · Authors · 2026-04-07
> > >
> > > Thank you for your follow-up. We apologize that our previous rebuttal did not fully address your concerns about stability and computational complexity. Below we provide additional clarification and new results (see attachment: https://anonymous.4open.science/r/HyperPotter-6C052325/r1.pdf).
> > > # 1) Stability analysis
> > >
> > > Following your comments, we assess “stability” in two complementary senses: (i) construction-level stability under variations of key hypergraph parameters, and (ii) training-level stability under different random seeds.
> > > ## 1.1 Stability w.r.t. hypergraph construction hyperparameters
> > >
> > > In the previous rebuttal, we reported sensitivity to the primary hypergraph parameters $K$ (hyperedge count) and $D$ (degree-related setting). Here, we additionally evaluate stability w.r.t. the maximum number of FCM iterations $T$, which directly controls the convergence depth of the per-sample clustering.
> > >
> > > We test $T\in$ {3, 5, 7} while keeping all other settings fixed. As shown in Table 16 (attachment), the default $T=5$ performs best overall, while $T=3$ or $T=7$ leads to only modest changes in EER. This indicates that hyperedge construction is not overly sensitive to the iteration budget, supporting construction-level stability.
> > > ## 1.2 Stability across random seeds
> > >
> > > We evaluate HyperPotter under three different random seeds to quantify robustness to stochastic initialization and optimization. (Due to time constraints, we report results for three seeds on a few representative datasets here, and can extend to more seeds if needed.)
> > >
> > > The results are listed in Table 17 (attachment) and are consistent with the trends reported in our paper when averaged across seeds. Across the tested datasets, HyperPotter exhibits low variance across seeds (std < 0.5% EER points on ITW/21LA/21DF datasets), and it consistently outperforms the baseline under the same seeding protocol.
> > >
> > > We believe the reported settings have considered the most critical hyperparameters for hypergraph construction and training stability. We would like to incorporate additional tests if the reviewer has further suggestions.
> > > # 2) Computional Complexibility Analysis
> > >
> > > We apologize that our earlier response focused on empirical costs (latency/throughput/VRAM) without clearly presenting an asymptotic complexity breakdown. Below we summarize both time and memory complexity.
> > > ## 2.1 Time complexity
> > >
> > > Let $N$ be the number of nodes, $F$ the feature dimension, $K$ the number of hyperedges, and $T$ the number of FCM iterations. The time complexity mainly consists of three parts:
> > > - Prototype-guided FCM. Each iteration computes the distances between $N$ node features and $K$ centroids, costing $O(NFK)$ per iteration. Over $T$ iterations, this is $O(TNFK)$.
> > > - Amplification operator construction. The structural term $A^{(c)}=UU^\top$ costs $O(N^2K)$. The feature term $A^{(f)}=X'X'^\top/\sqrt{F}$ costs $O(N^2F)$. Hence this step costs $O\left(N^2(K + F)\right)$.
> > > - Relational evidence propagation. $Z=AX'$ and subsequent projections multiply the $N\times N$ operator $A$ with node features, which costs $O(N^2F)$ per propagation step.
> > >
> > > Overall, the time complexity can be summarized as: $O\left(TNFK+N^2(K+F)\right)$.
> > > ## 2.2 Memory complexity
> > >
> > > The memory complexity is mainly governed by the tensors in the three stages above:
> > > - Storing $X'$: $O(NF)$, storing prototypes: $O(KF)$, storing memberships $U$: $O(NK)$.
> > > - Constructing $A^{(c)},A^{(f)},A\in\mathbb{R}^{N\times N}$: $O(N^2)$.
> > > - Storing propagated evidence $Z$: $O(NF)$.
> > >
> > > Overall, the memory complexity can be summarized as: $O\left(N^2+NK+NF+KF\right)$.
> > > ## 2.3 Practical implications & empirical validation
> > >
> > > - The computational complexity is primarily driven by $N$ via the $N^2$ components. In our setting, $N$ corresponds to the pooled node set produced by the graph builder (not raw frame-level tokens), so $N$ remains moderate in practice.
> > > - $K$ is an effective efficiency knob: it impacts both the FCM cost $O(TNFK)$ and the structural operator cost $O(N^2K)$.
> > >
> > > We empirically validate these trends with controlled sweeps over $N/K/T$ (Table 15, attachment). Among the tested factors, reducing $K$ provides the most favorable efficiency–accuracy trade-off. Decreasing $K$ from the default to $0.25N$ reduces latency (73.06→60.27 ms), improves throughput (25.81→38.52 utt/s), and lowers peak memory (14.63→13.76 GiB), while the mean EER changes only from about 6.27% to 6.74% (Table 3 in the paper).
> > >
> > > ---
> > >
> > > Overall, these results indicate that HyperPotter introduces a moderate and tunable computational overhead, and its stability is supported by both parameter sensitivity tests and multi-seed evaluation.
> > >
> > > We hope the new $T$ tests and multi-seed mean±std results address the stability concern directly, and the time/memory complexity analysis and empirical validation address the computational complexity concern. If these perspectives are still insufficient, please let us know and we will further expand them in the revised version.

---

### Decision · Program_Chairs · 2026-04-30

**Decision:**

Accept (regular)

**Comment:**

All reviewers agree to accept this paper.